# FUNCTIONAL SHRINKAGE FOR INPUT OPTIMIZATION

## ABSTRACT

Input optimization is a major methodology in deep learning that has been widely applied in adversarial attacks, neural network verification, and deep reinforcement learning. An important topic is how to inject a perturbation with a limited budget to the crucial part of a network, in order to avoid unstable optimization paths or meaningless outcomes. In this work, we formulate a network as a functional of activation functions, and propose a functional shrinkage method to extract the principal component of the network. Then the residual component can be vulnerable to perturbations, which is exploited for perturbation injection. Experimental results show that the proposed method achieves state-of-the-art performance in several input optimization tasks. It provides a new insight into the structural decomposition and weakness exploitation of neural networks.

## 1 INTRODUCTION

Input optimization is a different aspect from the traditional parameter optimization in deep learning (Erhan et al., 2009; Gurumurthy et al., 2021b; Yu & Gao, 2024). It intends to modify the input of a neural network (NN), in order to achieve some special objectives. Denote $x$ and $\theta$ as the input and model parameter for an NN $f$, respectively. Let $\mathcal{L}(f(x), \theta)$ be the loss function for the current machine learning task. Input optimization does not train $\theta$, but optimizes $x$ in some domain of interest $\mathcal{X}$ to maximize the loss:

$$x^* = \arg\max_{x \in \mathcal{X}} \mathcal{L}(f(x), \theta). \tag{1}$$

It originates from the idea of maximizing the activation of a given hidden unit $\eta_j^{(i)}$ by setting $\mathcal{L}(f(x), \theta) = \eta_j^{(i)}(x, \theta)$, which aims to find a good first-order representation of this unit (Erhan et al., 2009). Instead of exhaustively searching $\mathcal{X}$ for $x^*$, a simple **gradient ascent** step can make the input closer to the objective:

$$x_{(k+1)} = x_{(k)} + \zeta \nabla_x \mathcal{L}(f(x), \theta). \tag{2}$$

This forms the cornerstone for multiple research directions. Taking the adversarial attack as an example, it needs to attack the vulnerable part of an NN, in order to maximize the attack outcome with the lowest perturbation *budget* (Madry et al., 2017; Wong & Kolter, 2018; Ilyas et al., 2019; Zhang et al., 2022). Other examples include finding better actions in reinforcement learning (Lillicrap et al., 2015; Fujimoto et al., 2018; 2019), verifying NNs (Bunel et al., 2018; Cohen et al., 2019; Shi et al., 2023), input inversion (Dosovitskiy & Brox, 2016), and policy destabilization in reinforcement learning (Pinto et al., 2017).

Different from learning a large scale of model parameters effectively and efficiently, input optimization generally deals with lower-dimensional input features but more complex loss landscapes that even have combinatorial geometrical structures (Agrawal et al., 2019; Gurumurthy et al., 2021a). Recent works have found that activation functions can be continuously tuned in descent directions to construct more diversified perturbations (Hanin & Rolnick, 2019; Yu & Gao, 2024). This leads to the **alternating update scheme of both the input $x$ and the activation value $\eta$**:

$$x_{(k+1)} = x_{(k)} + \alpha_x \nabla_x \mathcal{L}(x_{(k)}, \eta_{(k)}), \ \eta_{(k+1)} = \eta_{(k)} + \alpha_\eta \nabla_\eta \mathcal{L}(x_{(k+1)}, \eta_{(k)}), \quad \alpha_x, \alpha_\eta \geqslant 0. \tag{3}$$

Compared with standard adversarial attack, current input optimization approach additionally optimizes the activation structure, which echoes its original motivation. This enables input optimization to be effectively applied to a wider arrays of machine learning tasks, such as neural policy learning (see Section 4.2) and deep reinforcement learning (see Section 4.3).

Although activation patterns have been enriched, it remains unsolved that which activation patterns are crucial to an effective input optimization, so that a perturbation with a limited budget can be injected to these crucial parts. In this work, we further formulate all the activation values $\{\eta_j^{(i)}\}$ as a holistic activation map $H(x)$, where the latter is no longer a *fixed value*, but a function of the input $x$. This enables a global, continuous, and functionally-aware decomposition of network behavior, as well as advanced mathematical tools to separate stable and fragile components of the NN. Specifically, a spectral decomposition (detailed in Theorem 3) yields:

$$H(x) = \underbrace{(\sigma_1 \Psi_1(x) + \cdots + \sigma_r \Psi_r(x))}_{\text{Principal Component}} + \underbrace{(\sigma_{r+1} \Psi_{r+1}(x) + \cdots + \sigma_R \Psi_R(x))}_{\text{Residual Component}}, \quad \sigma_1 \geqslant \cdots \geqslant \sigma_R \geqslant 0, \quad (4)$$

where $\sigma_r$ and $\Psi_r$ denote the $r$-th spectral value and component, respectively. Similar to the principal component analysis (PCA Vaswani et al. 2018), the principal component with high spectral values (Figure 1b) realizes the core function of the NN, while the residual component with low spectral values (Figure 1c) is an ambiguous operation and sensitive to noise and outliers. Hence the latter can be a good target to strengthen the effectiveness of input optimization, in order to relieve the problem of divergent or insignificant optimization (Goodfellow et al., 2015; Uesato et al., 2018; Athalye et al., 2018; Song et al., 2018). To do this, we propose a **Functional Shrinkage method for Input Optimization (FSIO)**, whose main contributions can be summarized as follows.

**1.** We formulate an NN functional as an activation map that lies in a Hilbert space with a well-defined inner product, where the structural information of the NN can be characterized and extracted.

**2.** The activation map can be decomposed into several orthogonal components, whose singular values represent their corresponding importance scores.

**3.** We develop a functional shrinkage algorithm to extract the principal component of the NN functional. Under a basic condition of local spectral stability, this algorithm converges weakly to an optimal solution with theoretical guarantees. We further exploit the residual component to construct a regularization term for input optimization.

The entire working pipeline and the step-by-step explanations of FSIO are shown in Figure 2. A significant difference between FSIO and mainstream adversarial attack methods is that the former **optimizes both the input and the activation map**, while the latter generally modify only the input.

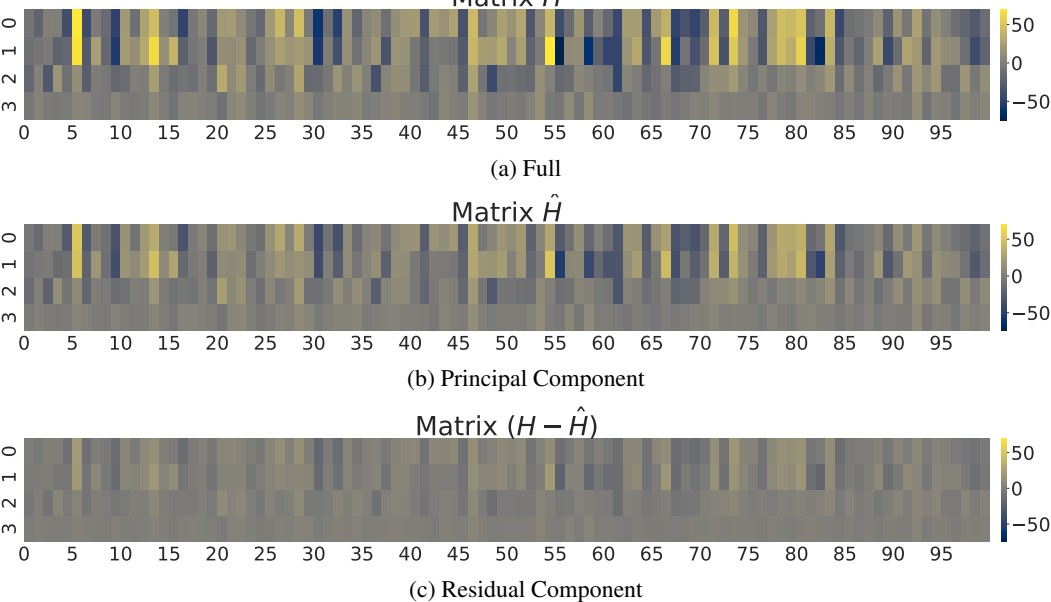

Figure 1: Activation maps extracted by the proposed FSIO method. (a) The full activation map of the entire neural network, which can be decomposed into the principal component and the residual component. (b) The principal component contains the main structural information and dominant signals, which form regular patterns. (c) The residual component contains the outlier-sensitive information and noise, which form random patterns.

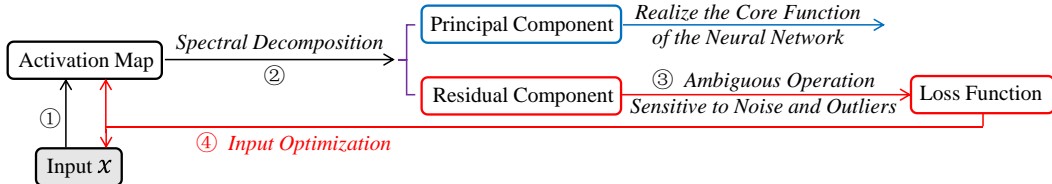

Figure 2: Diagram of the proposed Functional Shrinkage method for Input Optimization (FSIO). ① The input $x$ is processed to form the activation map. ② A spectral decomposition is implemented on the activation map to extract the principal and residual components. ③ The residual component is exploited to construct the loss function. ④ The loss function is used to **optimize both the input and the activation map**. This loop continues until both the input and activation map are sufficiently optimized.

## 2 PRELIMINARIES AND RELATED WORKS

We introduce some preliminaries and related works on adversarial attack, activation patterns, and shrinkage algorithms.

### 2.1 ADVERSARIAL ATTACK

Generally speaking, input optimization aims to degrade model performance by constructing adversarial examples that maximize the predictive loss under limited perturbation budgets (Szegedy et al., 2013; Carlini & Wagner, 2017). Adversarial attack is a main type of input optimization that focuses on inducing misclassifications by a classifier (Goodfellow et al., 2015; Madry et al., 2017; Wong & Kolter, 2018; Ilyas et al., 2019; Xu et al., 2020):

$$\max_{\|\delta\|_p \leqslant \epsilon} \mathcal{L}(f(x + \delta), y), \tag{5}$$

where $\mathcal{L}$ is the loss function, $f$ is the NN, $y$ is the ground-truth label (or response) for the input $x$ in supervised learning, sign denotes the element-wise sign function, and $\|\delta\|_p \leqslant \epsilon$ indicates that the perturbation budget $\|\delta\|_p$ based on the $\ell_p$ norm cannot exceed $\epsilon$. A popular attack strategy is Projected Gradient Descent (PGD, Madry et al. 2017; Croce & Hein 2019), which is the multi-iteration extension of Fast Gradient Sign Method (FGSM, Goodfellow et al. 2015):

$$\tilde{x}_{(k+1)} = \text{proj}_{\mathcal{B}_p[x;\epsilon]}(\tilde{x}_{(k)} + \zeta' \cdot \text{sign}(\nabla_x \mathcal{L}(f(\tilde{x}_{(k)}), y))), \quad k = 1, 2, \cdots, \tag{6}$$

where $\text{proj}_{\mathcal{B}_p[x;\epsilon]}$ denotes the projection operator onto the closed neighborhood of $x$ with an $\ell_p$-norm radius of $\epsilon$, and $\zeta' \geqslant 0$ denotes the step size. By implementing multiple iterations, PGD aims to achieve a stronger attack of "first-order adversary". However, there are still some unsolved challenges in adversarial attack regarding optimization stability and outcome reliability, such as unstable optimization trajectory (Nguyen et al., 2015; Szegedy et al., 2013), sensitivity to initialization (Eykholt et al., 2018; Shin & Song, 2020), sensitivity to local minima (Shin & Song, 2022), and existence of multiple solutions (Fawzi et al., 2018). They indicate that optimizing only the input $x$ may be insufficient for broader applications, and other parts of the NN should also be optimized simultaneously.

### 2.2 ACTIVATION-DESCENT REGULARIZATION

To address the above challenges, Yu & Gao (2024) propose the Activation-Descent Regularization (ADR-GD) method to enrich activation patterns for ReLU networks. By introducing continuous activation variables, ADR-GD transforms the original discrete activation patterns into differentiable representations, and incorporates regularization terms to align input updates with the descent directions in the activation space. This approach improves attack performance over traditional methods like FGSM and PGD, especially around activation boundaries. Its core regularization term is defined as follows:

$$\mathcal{L}_i(x, \eta) := \sum_{j=1}^{d_i} \left( \left( \frac{h_j^{(i)}(x)}{\|P_j^{(i)}\|_2} \cdot \left( \frac{1}{2} - \eta_j^{(i)} \right)_+ \right)_+ + \left( -\frac{h_j^{(i)}(x)}{\|P_j^{(i)}\|_2} \cdot \left( \eta_j^{(i)} - \frac{1}{2} \right)_+ \right)_+ \right),$$
$$i \in [l] := \{1, 2, \cdots, l\}, \tag{7}$$

where $h_j^{(i)}(x)$ and $\eta_j^{(i)}$ denote the affine function and the activation variable for the $j$-th neuron of the $i$-th layer, respectively. $(\cdot)_+ := \max\{0, \cdot\}$ denotes the positive part operator, and $P^{(i)}$ denotes the hyperplane normal vectors for the neurons in the $i$-th layer:

$$P^{(i)} = W_i \cdot \prod_{j=-(i-1)}^{-1} \left( \text{diag}(\phi_\alpha(\eta^{(-j)})) \cdot W_{-j} \right). \tag{8}$$

Without loss of generality, we directly present the surrogate network form as follows:

$$\begin{aligned}
\bar{h}^{(1)}(x, \eta) &:= W_1 \cdot x + b_1, \\
\bar{h}^{(i+1)}(x, \eta) &:= W_{i+1} \cdot \text{diag}(\phi_\alpha(\eta^{(i)})) \cdot \bar{h}^{(i)}(x, \eta) + b_{i+1}, \quad i \in [l], \\
\bar{f}(x, \eta) &:= \bar{h}^{(l+1)}(x, \eta),
\end{aligned} \tag{9}$$

where $\phi_\alpha(\eta^{(i)}) := (1 + \exp(-\alpha(\eta^{(i)} - \frac{1}{2})))^{-1}$ defines the sigmoid function with an offset of $\frac{1}{2}$. Note that $\phi_\alpha$ is an element-wise operator on the vector $\eta^{(i)} \in \mathbb{R}^{d_i}$, and $\text{diag}$ denotes the diagonalization operator. $W_{i+1} \in \mathbb{R}^{d_{i+1} \times d_i}$ denotes the weight matrix that connects the $i$-th layer to the $(i+1)$-th layer, where $d_i$ and $d_{i+1}$ denote the dimensionalities for these two layers, respectively. In addition, $\eta_j^{(i)}$ should satisfy the following feasibility constraints in the optimization:

$\eta_j^{(i)} : \begin{cases} \geq \frac{1}{2}, & \text{if } h_j^{(i)}(x) \geq 0, \\ < \frac{1}{2}, & \text{otherwise.} \end{cases}$. On the other hand, we use the notation without a bar

$$h^{(i+1)}(x, \vartheta) := W_{i+1} \cdot \text{diag}(\vartheta^{(i)}) \cdot h^{(i)}(x, \vartheta) + b_{i+1}, \ i \in [l], \ \vartheta_j^{(i)} = \begin{cases} 1, & \text{if } h_j^{(i)}(x) \geq 0, \\ 0, & \text{otherwise.} \end{cases} \tag{10}$$

to represent the original (non-surrogate) network, where $\vartheta$ is the original activation function. ADR-GD achieves effective performance in ReLU based NNs, but loses some advantage in NNs with other activation functions (see Table 1). **It indicates that only enriching activation patterns may not be robust enough for input optimization. Further fundamental approaches regarding the structure of an NN may be needed for substantial improvements.**

## 2.3 SHRINKAGE METHODS

In various areas of machine learning, it is widely recognized that a function, signal, or system can be largely represented by several principal components, whereas the residual components contain the outlier-sensitive information and noise. This concept motivates a bunch of useful applications, including PCA (Vaswani et al., 2018), sparse learning (Lai & Yang, 2025; Lin et al., 2024a;b), compressed sensing (Wang et al., 2025; Lai & Wang, 2024), and operations research (Lai et al., 2025; 2024). A typical shrinkage method is the singular value thresholding (SVT) algorithm (Cai et al., 2010), which can be applied in the low-rank matrix completion:

$$\min_{X \in \mathbb{R}^{N \times d}} \|X\|_* \quad \text{s.t.} \quad \text{proj}_\Omega(X) = \text{proj}_\Omega(M), \tag{11}$$

where $\|\cdot\|_*$ denotes the nuclear norm, $\text{proj}_\Omega$ denotes the projection operator onto the span of matrices vanishing outside the index set $\Omega$, and $M$ denotes the observed signal to be recovered. The corresponding shrinkage method is:

$$\begin{cases} X_{(k)} = \text{shrink}(X'_{(k-1)}, \tau) := U_{(k-1)}(\Sigma_{(k-1)} - \tau I)V_{(k-1)}^\top, \\ X'_{(k)} = X'_{(k-1)} + \varsigma_k \text{proj}_\Omega(M - X_{(k)}), \end{cases} \tag{12}$$

where $X'_{(k-1)} := U_{(k-1)}\Sigma_{(k-1)}V_{(k-1)}^\top$ is a singular value decomposition (SVD) of $X'_{(k-1)}$, and $I$ denotes the identity matrix. Hence this shrinkage method actually shrinks the singular values $\Sigma_{(k-1)}$ towards 0, in order to reduce its nuclear norm and produce a low-rank recovery (the nuclear norm approximates the matrix rank). This technique can be extended to decompose and adjust the activation map of an NN, which improves the resulted input optimization.

## 3 METHODOLOGY

Based on the above analysis, we can formulate an NN as a functional of activation functions, then shrink this functional to extract the main principal component of the NN. The resulted residual component can be vulnerable to perturbations and noise, which can be exploited for input optimization.

### 3.1 ACTIVATION MAP

We propose a kind of activation map as a framework of the NN functional. In a general NN structure (9), there are $l$ layers in total and $d_i$ neurons for the $i$-th layer with $i \in [l]$. Let $d_{max}$ and $d_{min}$ be the largest and the smallest numbers of neurons among these layers. We recall that $\eta_j^{(i)}$ is the activation variable for the $j$-th neuron of the $i$-th layer. In our formulation, $\eta_j^{(i)}$ is not just a value or one-layer activation, but a composite activation function that directly maps the input $x$ to the current activation value $\eta_j^{(i)} : \mathcal{X} \mapsto [0, 1]$. Without loss of generality, we can assume that $\eta_j^{(i)} \in L^2(\mathcal{X})$ (the square-integrable space) and the domain $\mathcal{X} \subseteq \mathbb{R}^n$ is compact. Then we can define the activation map of the entire network as follows:

$$\eta^{(i)}(x) := \underbrace{[\eta_1^{(i)}(x), \eta_2^{(i)}(x), \cdots, \eta_{d_i}^{(i)}(x), 0, \cdots, 0]}_{d_{max}} \in (L^2(\mathcal{X}))^{1 \times d_{max}}, \quad i \in [l],$$

$$H'(x) := [\eta^{(1)}(x); \eta^{(2)}(x); \cdots; \eta^{(l)}(x)] \in (L^2(\mathcal{X}))^{l \times d_{max}},$$

$$H(x) := H'(x) \cdot W_{red} =: [\mu_j^{(i)}(x)] \in (L^2(\mathcal{X}))^{l \times d_{min}}, \tag{13}$$

where each layer $\eta^{(i)}(x)$ is padded with 0 to form a $d_{max}$-dimensional row vector of functions, and the initial activation map $H'(x)$ is an $l \times d_{max}$-dimensional matrix of functions. Similar to the autoencoder (Ackley et al., 1985), the valid representation dimensionality of an NN is determined by the lowest layer dimensionality $d_{min}$. Thus we use a full-rank matrix $W_{red} \in \mathbb{R}^{d_{max} \times d_{min}}$ (which can be randomly generated) to transform $H'(x)$ into a compact activation map $H(x)$. First, we present an algebraic property of $H(x)$.

**Theorem 1.** *The rank of $H(x)$ is never larger than $\min\{l, d_{min}\}$ when $x$ changes in $\mathcal{X}$.*

The proof is given in Appendix A.1.

Next, we develop an inner product and establish a Hilbert space $\mathcal{H}$ for the activation map $H(x)$.

**Theorem 2.** *The following inner product is well-defined for $\mathcal{H} := (L^2(\mathcal{X}))^{l \times d_{min}}$ to be a Hilbert space.*

$$\langle H, G \rangle := \int_{\mathcal{X}} \mathrm{tr}(H(x)^\top G(x)) \, d\nu(x), \quad \forall H, G \in \mathcal{H}, \tag{14}$$

*where $\mathrm{tr}(\cdot)$ denotes the trace operator, and $\nu$ denotes a measure on $\mathcal{X}$ for the function space $L^2(\mathcal{X})$. The induced norm for $\mathcal{H}$ is denoted by $\|H\|_{\mathcal{H}} := \sqrt{\langle H, H \rangle}$.*

The proof is given in Appendix A.2. This inner product first aggregates all the elements $\sum_{i=1}^{l} \sum_{j=1}^{d_{min}} [H(x) \odot G(x)]_{ij}$, then integrates this sum on $\mathcal{X}$ with the measure $\nu$, where $\odot$ denotes the element-wise multiplication. $\nu$ can be chosen according to different tasks. For example, $\nu$ can be the Lebesgue measure for general applications. If $\nu$ is set as the Dirac delta function on $x$, then $\langle H, G \rangle$ reduces to $\mathrm{tr}(H(x)^\top G(x))$ and the induced norm is the Frobenius norm $\|H(x)\|_F$.

### 3.2 SINGULAR VALUE DECOMPOSITION OF ACTIVATION MAP

SVD is a common approach to decompose a matrix into several orthonormal components for further applications, such as feature extraction, denoising, signal compression, etc. However, an activation map $H(x)$ is a matrix of functions instead of a matrix of fixed values. Hence it is nontrivial to develop an SVD scheme for $H(x)$. We observe that an input optimization task involves only a small neighborhood $\mathcal{X} := \mathcal{B}_p[x; \epsilon]$ of a fixed input $x$, due to a limited pertabation budget $\epsilon$. Thus it is tractable to extend the traditional SVD to establish an orthogonal decomposition of $H(x)$ as follows:

**Theorem 3.** *Given any fixed input $x$, let the domain $\mathcal{X} := \mathcal{B}_p[x; \epsilon]$. Then given any $H \in \mathcal{H}$, there exist an $R \leqslant \min\{l, d_{min}\}$, a set of singular values $\{\sigma_r\}_{r=1}^R$, and a set of orthogonal basis $\{\Psi_r(z) \in \mathcal{H}\}_{r=1}^R$ such that*

$$H(z) := \sum_{r=1}^{R} \sigma_r \Psi_r(z), \quad \sigma_1 \geqslant \sigma_2 \geqslant \cdots \geqslant \sigma_R \geqslant 0, \quad \forall z \in \mathcal{B}_p[x; \epsilon], \tag{15}$$

*where the first $R'$ ($R' \leqslant R$) singular values $\{\sigma_r\}_{r=1}^{R'}$ are the essential singular values of the matrix $H(x)$. Note that $\{\sigma_r\}_{r=1}^R$ and $\{\Psi_r(z)\}_{r=1}^R$ are dependent on $x$, but we omit such notations since the domain $\mathcal{B}_p[x; \epsilon]$ is already $x$-specified.*

The proof is given in Appendix A.3, while we provide a sketch here. We can first conduct the SVD of $H(x)$ to obtain the initial values of $\{\sigma_r\}_{r=1}^R$ and $\{\Psi_r(x)\}_{r=1}^R$, then extend $\{\Psi_r(x)\}_{r=1}^R$ to $\{\Psi_r(z)\}_{r=1}^R$ for any $z \in \mathcal{B}_p[x; \epsilon]$. Note that $\{\Psi_r(z)\}_{r=1}^R$ are mutually orthogonal but not necessarily normal. They are necessarily normal only at $z = x$.

Theorem 3 characterizes the activation map $H$ in the neighborhood $\mathcal{B}_p[x; \epsilon]$. $H$ can be decomposed into several orthogonal components $\{\Psi_r\}_{r=1}^R$, whose singular values $\{\sigma_r\}_{r=1}^R$ represent their corresponding importance scores. The orthogonality of $\{\Psi_r\}_{r=1}^R$ ensures that the signals conveyed by different components are uncorrelated, which facilitates the principal component extraction in Section 3.3. $\{\sigma_r\}_{r=1}^R$ are calculated at the current input $x$, and dominate any $z \in \mathcal{B}_p[x; \epsilon]$, which covers all the possible attacked inputs. Therefore, by letting $z = x + \delta$ with a perturbation $\delta$, (15) is applicable to input optimization. This finding enables us to develop a functional shrinkage method that extracts a holistic principal component with respect to (w.r.t.) the entire perturbation region.

### 3.3 FUNCTIONAL SHRINKAGE

The next stage is to extract the principal component of $H$ based on the decomposition in (15). We develop a functional shrinkage method that shrinks the singular values $\{\sigma_r\}_{r=1}^R$ towards 0, in order to uncover this principal component. First, we introduce the concept of differential for a functional $\mathfrak{f} : \mathcal{H} \mapsto \mathbb{R}$, which is analog to the gradient of a function. More details can be found in a textbook of variational analysis (Rockafellar & Wets, 2009).

**Definition 4** (The Fréchet Differential). The Fréchet differential $\nabla\mathfrak{f}(H)$ is some element $G \in \mathcal{H}$ that satisfies the following criterion. It is unique if it exists.

$$\lim_{\substack{\bar{H} \xrightarrow{\mathcal{H}} H \\ \bar{H} \neq H}} \frac{\mathfrak{f}(\bar{H}) - \mathfrak{f}(H) - \langle G, \bar{H} - H \rangle}{\|\bar{H} - H\|_{\mathcal{H}}} = 0. \tag{16}$$

Next, we investigate the following two functionals that will be used later.

**Theorem 5.** *(1) The functional $\mathfrak{f}(H) := \frac{1}{2}\|H\|_{\mathcal{H}}^2$ is convex w.r.t. $H$, and its Fréchet differential is $\nabla\mathfrak{f}(H) = H$.*

*(2) The functional $\mathfrak{g}(H) := \sum_{r=1}^{R'} \sigma_r(x)$ is convex w.r.t. $H$, where $x$ is the given input in Section 3.2 and $R'$ is the essential rank of $H(x)$ defined in Theorem 3.*

The proof is given in Appendix A.4. Next, we introduce the proximal mapping w.r.t. $\mathfrak{g}(\cdot)$ as follows:

$$\mathrm{prox}_{\tau\mathfrak{g}(\cdot)}(H) := \underset{G \in \mathcal{H}}{\arg\min} \left\{ \frac{1}{2}\|G - H\|_{\mathcal{H}}^2 + \tau\mathfrak{g}(G) \right\}, \quad \tau \geqslant 0. \tag{17}$$

It aims to find a function $G$ that not only approximates $H$ but also has a relatively low spectral energy $\mathfrak{g}(\cdot)$. Similar to the matrix case (Cai et al., 2010), $\mathrm{prox}_{\tau\mathfrak{g}(\cdot)}(H)$ has the following closed-form solution:

**Theorem 6.** *Following the SVD of $H$ in (15),*

$$\mathrm{prox}_{\tau\mathfrak{g}(\cdot)}(H) = \sum_{r=1}^R (\sigma_r - \tau)_+ \Psi_r. \tag{18}$$

The proof is given in Appendix A.5. To reduce $\mathfrak{g}(H)$, the singular values are uniformly reduced by the same step size $\tau$, then the small ones will shrink to 0 and the rank will decrease accordingly. This mechanism results in a low-rank recovery of $H$, which contains the main structural information and dominant signals of $H$.

Now we can present the optimization model that extracts the principal component $\hat{H}$:

$$\hat{H} := \underset{G \in \mathcal{H}}{\arg\min} \left\{ \mathfrak{h}(G) := \frac{1}{2}\|\mathrm{proj}_\Omega(G - H)\|_{\mathcal{H}}^2 + \tau\mathfrak{g}(G) \right\}, \quad \tau \geqslant 0, \tag{19}$$

where $\mathrm{proj}_\Omega$ serves as a mask that only keeps the concerned index set $\Omega$ of $(G - H)$ and sets the other elements as zeros. It provides a more flexible scheme for practical use, especially when we can only observe or pay attention to some particular activation patterns, which is like the attention mechanism

(Vaswani et al., 2017). We develop a functional shrinkage algorithm based on the proximal forward-backward splitting criterion (Combettes & Wajs, 2005; Cai et al., 2010) as follows:

$$G_{(k+1)} = \text{prox}_{\varsigma\tau\mathfrak{g}(\cdot)}(G_{(k)} + \varsigma\text{proj}_\Omega(H - G_{(k)})), \tag{20}$$

where $\varsigma > 0$ is the learning rate. More details can be found in Appendix A.6. To obtain a solution to (19), the following condition about local spectral stability is needed. It means that the matrix nuclear norm of $H(z)$ should be stable in a neighborhood of $x$, while this neighborhood can be arbitrarily small. It is a basic and reasonable condition that can be satisfied in real-world scenarios.

**Condition 1** (Local Spectral Stability). Following the SVD of $H$ in (15), there exists another neighborhood $\mathcal{B}_p[x; \epsilon'] \subseteq \mathcal{B}_p[x; \epsilon]$ such that the matrix nuclear norm $\|H(z)\|_* := \sum_{r=1}^R \sigma_r(z)$ is a continuous function on $\mathcal{B}_p[x; \epsilon']$.

**Theorem 7** (Convergence Theorem). *If $H$ satisfies Condition 1 and $\varsigma \in (0,1)$, then the sequence generated by (20) converges weakly to a solution $G^*$ to (19). Moreover, the objective value sequence $\{\mathfrak{h}(G_{(k)})\}_{k=1}^\infty$ is non-increasing and converges to $\mathfrak{h}(G^*)$ with a sublinear rate $\mathcal{O}(\frac{1}{k})$.*

The proof is given in Appendix A.6. This algorithm is a kind of fixed-point algorithm and the convergence is based on the nonexpansivity of the iterative operator. It generally obtains only weak convergence due to the complexity of the function space $L^2(\mathcal{X})$, but it is fully capable for practical use. Last, the principal component $\hat{H}$ is transformed back into the original shape by $\hat{H}' := \hat{H} \cdot (W_{red}^\top W_{red})^{-1} \cdot W_{red}^\top \in (L^2(\mathcal{X}))^{l \times d_{max}}$, whose $ij$-th element corresponds to the recovered activation variable $\hat{\eta}_j^{(i)}$. Then the residual component $(H' - \hat{H}')$ contains outlier-sensitive information and noise, which is vulnerable to adversarial attacks and thus can be exploited for input optimization.

## 3.4 OPTIMIZATION MODEL

In the last stage, we develop an appropriate regularization term with the residual component $(H' - \hat{H}')$ and the corresponding objective function as follows:

$$S_i(x, H', \hat{H}') := \sum_{j=1}^{d_{max}} \exp\left(-\lambda_0 \frac{|h_j^{(i)}(x)|}{\|P_j^{(i)}\|}\right)(\hat{H}_{ij} - \hat{H}'_{ij})^2, \tag{21}$$

$$L^*(x, H', \hat{H}') = -J(\bar{h}^{(l+1)}(x, H')) + \sum_{i=1}^l \lambda_1(\mathcal{L}_i(x, H') + S_i(x, H', \hat{H}')), \tag{22}$$

where $\lambda_1 \geqslant 0$ controls the strengths for the activation-descent regularization (7) and the residual regularization (21). $\lambda_0 > 0$ is a hyperparameter that adjusts the shape of the exponential function, while $J$ is the pure objective that should be *maximized* in a particular machine learning task. In $S_i(x, H', \hat{H}')$, each summand (corresponding to each neuron) is determined by the activation strength $|h_j^{(i)}(x)|$ and the residual magnitude $(\hat{H}_{ij} - \hat{H}'_{ij})^2$. In brief, a neuron with higher residual magnitude but smaller activation strength takes up higher regularization strength. Then this neuron will receive more intensive adversarial attacks in the process of minimizing $L^*$, until it gets more activated or lowers residual magnitude. By this means, the perturbed input dynamically adjusts itself to activate the residual component. An adversarial training scheme similar to that of (Yu & Gao, 2024) is used for the input and other variables:

$$\begin{cases} x \leftarrow x - \alpha_x \nabla_x L^*, \quad H' \leftarrow H' - \alpha_{H'} \partial_{H'} L^*, \\ \lambda_1 \leftarrow \lambda_1 + \alpha_{\lambda_1} \nabla_{\lambda_1} L^*, \end{cases} \tag{23}$$

where $x$ and $H'$ use the (sub)gradient descent update (primal), while $\lambda_1$ uses the gradient ascent update (dual). $(\alpha_x, \alpha_{H'}, \alpha_{\lambda_1})$ are the corresponding nonnegative step sizes.

## 4 EXPERIMENTAL RESULTS

We extend the evaluating framework of (Yu & Gao, 2024) to include NNs with different activation functions, including ReLU, softplus, and sigmoid. Four adversarial attack approaches are taken into comparisons: FGSM, PGD, APGD, AutoAttack (Croce & Hein, 2020), ADR-GD, and C&W

(Carlini & Wagner, 2017). While the perturbation budgets $\epsilon$ for other competitors are set according to different tasks and data sets, the $\epsilon$ of C&W is automatically set by its own scheme. 1000 attack iterations are used for PGD, APGD, the APGD wrapped in AutoAttack, and FSIO, while the attack iterations used for C&W are automatically set by itself. Three mainstream data sets are used in the experiments: MNIST (Deng, 2012), CIFAR10 (Krizhevsky, 2009), and ImageNet (Deng et al., 2009). Experimental settings are provided in Appendix A.7, while the ablation study is provided in Appendix A.8.

Table 1: Attack success rates (%, upper) and optimized objective values (lower) of different methods on benchmark data sets. The left nine columns are for untargeted attacks and the right three columns are for targeted attacks. A higher score indicates better performance in this task. The mean and standard deviation (STD) results of each case are reported.

| Method | | MNIST ReLU | MNIST Softplus | MNIST Sigmoid | CIFAR-10 ReLU | CIFAR-10 Softplus | CIFAR-10 Sigmoid | ImageNet ReLU | ImageNet Softplus | ImageNet Sigmoid | MNIST Targeted | CIFAR-10 Targeted | ImageNet Targeted |
|---|---|---|---|---|---|---|---|---|---|---|---|---|---|
| FGSM | MEAN | 51.82 | 52.73 | 50.00 | 68.18 | 67.27 | 59.09 | 36.36 | 33.64 | 34.55 | 16.36 | 31.81 | 26.36 |
| | STD | 5.12 | 5.34 | 4.98 | 5.87 | 5.21 | 4.79 | 3.92 | 3.65 | 3.43 | 4.11 | 3.98 | 4.05 |
| PGD | MEAN | 82.73 | 77.27 | 77.27 | 83.64 | 81.82 | 77.27 | 65.45 | 66.36 | 61.82 | 28.18 | 43.64 | 38.18 |
| | STD | 4.89 | 5.12 | 4.75 | 5.03 | 4.88 | 4.55 | 3.78 | 3.92 | 3.51 | 4.05 | 3.77 | 3.65 |
| APGD | MEAN | 83.34 | 78.33 | 77.62 | 84.42 | 82.55 | 78.09 | 67.00 | 68.28 | 63.29 | 28.44 | 44.14 | 38.93 |
| | STD | 4.95 | 5.16 | 4.80 | 5.06 | 4.91 | 4.59 | 3.88 | 4.04 | 3.57 | 4.09 | 3.82 | 3.72 |
| AutoAttack | MEAN | 85.85 | 79.19 | 79.28 | 84.28 | 83.69 | 78.61 | 70.40 | 69.47 | 63.14 | 28.66 | 44.06 | 39.06 |
| | STD | 4.96 | 5.21 | 4.81 | 5.07 | 4.97 | 4.69 | 3.92 | 4.09 | 3.65 | 4.13 | 3.81 | 3.85 |
| C&W | MEAN | 84.54 | 82.72 | 80.00 | 82.73 | 84.54 | 77.27 | 61.82 | 62.73 | 62.73 | 31.82 | 39.09 | 37.27 |
| | STD | 5.11 | 5.01 | 4.88 | 4.35 | 4.12 | 4.09 | 3.07 | 2.99 | 3.53 | 6.11 | 7.13 | 6.96 |
| ADR-GD | MEAN | 89.09 | 86.36 | 81.82 | 81.82 | 82.73 | 72.73 | 64.55 | 63.64 | 68.18 | 53.64 | 62.73 | 43.64 |
| | STD | 4.23 | 4.78 | 4.11 | 4.85 | 4.39 | 3.98 | 3.45 | 3.71 | 3.22 | 3.68 | 3.55 | 3.49 |
| **FSIO (ours)** | MEAN | **98.18** | **98.18** | **96.36** | **99.09** | **95.45** | **93.64** | **82.73** | **86.36** | **87.27** | **61.81** | **90.91** | **52.73** |
| | STD | **3.72** | **3.34** | **4.81** | **3.85** | **3.38** | **3.27** | **4.05** | **4.18** | **4.02** | **3.41** | **3.25** | **3.18** |

| Method | | MNIST ReLU | MNIST Softplus | MNIST Sigmoid | CIFAR-10 ReLU | CIFAR-10 Softplus | CIFAR-10 Sigmoid | ImageNet ReLU | ImageNet Softplus | ImageNet Sigmoid | MNIST Targeted | CIFAR-10 Targeted | ImageNet Targeted |
|---|---|---|---|---|---|---|---|---|---|---|---|---|---|
| FGSM | MEAN | 3.721 | 3.698 | 3.545 | 1.880 | 1.891 | 1.341 | 2.010 | 1.980 | 1.783 | -18.245 | -8.689 | -16.655 |
| | STD | 1.702 | 1.476 | 1.547 | 1.720 | 1.825 | 1.880 | 1.101 | 0.820 | 0.675 | 10.612 | 4.590 | 12.521 |
| PGD | MEAN | 7.972 | 7.175 | 7.833 | 2.845 | 2.574 | 2.383 | 5.233 | 5.004 | 4.938 | -22.120 | -7.620 | -12.638 |
| | STD | 3.011 | 3.332 | 2.803 | 2.280 | 2.171 | 1.991 | 1.995 | 2.012 | 1.630 | 15.544 | 6.478 | 11.412 |
| APGD | MEAN | 8.070 | 7.163 | 7.921 | 2.866 | 2.645 | 2.390 | 5.393 | 5.157 | 5.047 | -21.926 | -7.497 | -12.440 |
| | STD | 3.068 | 3.379 | 2.812 | 2.306 | 2.222 | 2.040 | 2.040 | 2.055 | 1.692 | 15.409 | 6.252 | 11.337 |
| AutoAttack | MEAN | 8.185 | 7.394 | 8.018 | 2.939 | 2.579 | 2.316 | 5.507 | 5.392 | 5.146 | -21.345 | -7.137 | -12.048 |
| | STD | 3.118 | 3.379 | 2.821 | 2.281 | 2.184 | 2.075 | 2.048 | 2.019 | 1.693 | 14.997 | 6.197 | 11.097 |
| C&W | MEAN | 0.866 | 0.971 | 0.731 | 0.258 | 0.308 | 0.280 | 0.503 | 0.554 | 0.499 | -21.763 | -8.397 | -11.003 |
| | STD | 0.610 | 0.492 | 0.509 | 0.319 | 0.289 | 0.291 | 0.195 | 0.102 | 0.136 | 16.275 | 5.537 | 10.776 |
| ADR-GD | MEAN | 8.941 | 8.734 | 8.960 | 3.217 | 2.811 | 2.699 | 6.554 | 6.412 | 6.399 | -7.198 | -4.042 | -7.214 |
| | STD | 3.141 | 3.995 | 3.122 | 1.582 | 1.493 | 1.759 | 1.850 | 1.711 | 1.108 | 1.023 | 1.005 | 0.988 |
| **FSIO (ours)** | MEAN | **12.455** | **11.285** | **13.775** | **4.315** | **3.244** | **3.345** | **7.010** | **6.710** | **7.516** | **-3.210** | **-1.644** | **-5.998** |
| | STD | **4.204** | **4.626** | **2.851** | **3.412** | **3.402** | **2.997** | **1.456** | **1.194** | **1.309** | **3.501** | **1.008** | **0.992** |

## 4.1 Adversarial Attacks on Image Classification

The adversarial attack framework in Section 2.1 is adopted to conduct image classification experiments. Adversarial examples targeting a specific neural image classifier are generated by different methods, and then fed into this classifier to examine the attack performance. We use a small-sized convolutional neural network for MNIST, the VGG16 (Sengupta & Shah, 2019) for CIFAR10, and the VGG19 (Sengupta & Shah, 2019) for ImageNet. The perturbation budgets $\epsilon$ for these three data sets are set to 0.1, 8/255, and 2/255, respectively. Moreover, we also examine the attack performance for different activation functions (i.e., ReLU, softplus, and sigmoid) of the fully-connected layers from the above NNs, in order to evaluate the extendability and robustness of different methods. Table 1 presents the attack success rates and optimized objective values of different methods, respectively. The proposed FSIO method outperforms other competitors in all the cases. For example, its attack success rates are approximately 10% to 40% higher than those of other competitors.

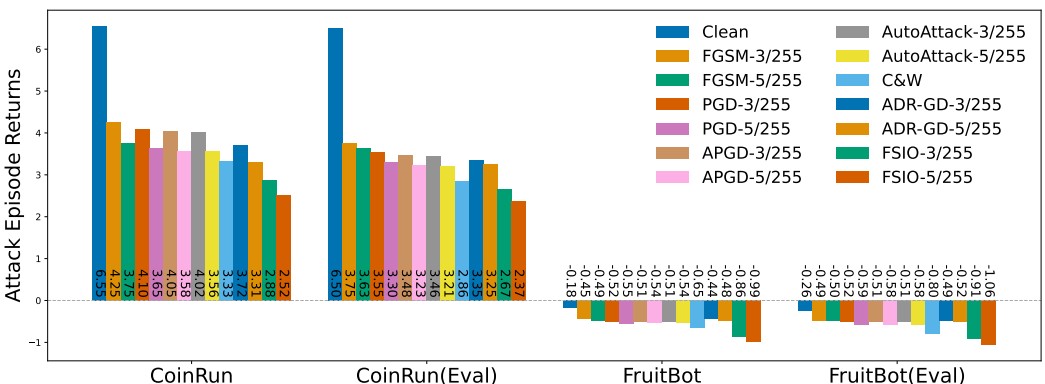

Figure 3: Adversarial attack results of different methods on neural policies with two levels of perturbation budgets $\epsilon = 3/255$ and $5/255$. A lower score indicates better performance in this task.

Moreover, FSIO achieves stable performance in all three kinds of activation functions, while other competitors deteriorate in some of them. These results indicate that FSIO is effective and robust in adversarial attacks on image classification.

### 4.2 ADVERSARIAL ATTACKS ON NEURAL POLICIES

The adversarial attack framework in Section 2.1 can also be used to attack neural policies. Following (Oikarinen et al., 2021; Yu & Gao, 2024), we use the environments that have an imagery observation space and a discrete action space. The Proximal Policy Optimization (PPO, Schulman et al. 2017) algorithm is used in this task. Different methods are used to attack the observation space, in order to yield non-optimal actions. Experiments are conducted on two Procgen environments (Cobbe et al., 2020): FruitBot and CoinRun, with two levels of perturbation budgets $\epsilon = 3/255$ and $5/255$. Results in Figure 3 show that FSIO with the smaller budget $3/255$ even outperforms other competitors with the larger budget $5/255$ in all the cases. It indicates that FSIO is effective in attacking neural policies.

### 4.3 INPUT OPTIMIZATION FOR DEEP REINFORCEMENT LEARNING

Input optimization can be used for refining inputs in deep reinforcement learning (DRL). Two DRL algorithms based on the actor-critic structure are considered: Deep Deterministic Policy Gradient (DDPG, Lillicrap et al. 2015) and Twin Delayed Deep Deterministic Policy Gradient (TD3, Fujimoto et al. 2018). In both algorithms, the actor model aims to capture the actions that maximize the Q-value approximated by the critic model. Specifically, the actor model $\pi : \mathbb{R}^n \mapsto \mathbb{R}^m$ maps an observed state to an action, while the critic model $V : \mathbb{R}^{n+m} \mapsto \mathbb{R}$ approximates the optimal Q-value from a state-action pair. Then an input optimization method is used to produce a perturbation vector $\delta \in \mathbb{R}^m$ that helps the actor capture the optimal actions that lead to near-optimal Q-values. It leads to the following exploitation action $a$ for a state $s \in \mathbb{R}^n$ and a probability threshold $\mathfrak{p} \in [0, 1]$:

$$a = \begin{cases} \pi(s), & \text{with probability } (1 - \mathfrak{p}), \\ \pi(s) + \underset{\delta \in \mathbb{R}^m}{\arg\max} V(s, \pi(s) + \delta), & \text{with probability } \mathfrak{p}. \end{cases} \tag{24}$$

We apply different input optimization methods to both DDPG and TD3 in three mujoco environments (Todorov et al., 2012) with $\mathfrak{p} = 0.25$. Results in Figure 4 show that FSIO outperforms other competitors in most cases, especially in the late stage of training. Hence FSIO is effective in input optimization for DRL.

## 5 CONCLUSION

It has been recognized that the activation patterns of a neural network (NN) play an important role in input optimization for deep learning. Although some recent works propose to enrich these activation patterns by introducing regularization terms that produce diversified landscapes of the loss function,

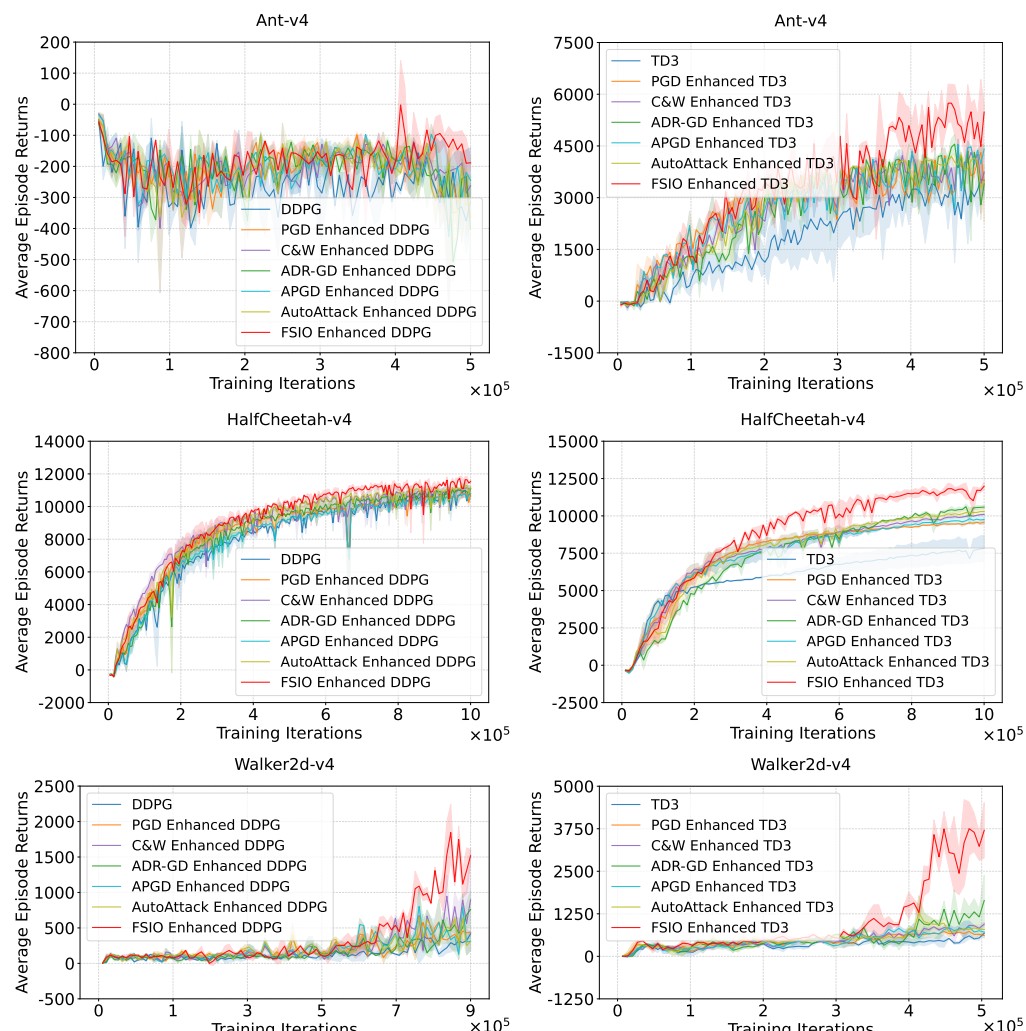

Figure 4: Input optimization for deep reinforcement learning with two algorithms (DDPG and TD3) in three mujoco environments (Ant-v4, HalfCheetah-v4, and Walker2d-v4). A higher score indicates better performance in this task.

they seldom dig into the inner structure to extract the crucial part of an NN. In this work, we propose a functional shrinkage method (FSIO) to decompose an NN into a principal component and a residual component. The former contains the main structural information and dominant signals, while the latter contains the outlier-sensitive information and noise. Thus the latter can be exploited for adversarial attacks and input optimization given a limited perturbation budget. Under a basic condition of local spectral stability, FSIO converges weakly to an optimal principal component with theoretical guarantees.

Experimental results show that FSIO outperforms other state-of-the-art methods in most cases of three input optimization tasks: adversarial attacks on image classification, adversarial attacks on neural policies, and input optimization for deep reinforcement learning. In the second task, FSIO with a smaller perturbation budget even outperforms other competitors with a larger perturbation budget. These results indicate that FSIO is effective in capturing the vulnerable part of an NN and injecting perturbations to this part. It provides a new insight into the structural decomposition and weakness exploitation of NNs. Future work may lie in extending this methodology to other deep learning tasks or reducing redundancy in deep neural architectures.

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

# A APPENDIX

## A.1 PROOF OF THEOREM 1

*Proof.* Given any fixed $x \in \mathcal{X}$, $H(x)$ is an ordinary $l \times d_{min}$-dimensional matrix, whose rank is no larger than $\min\{l, d_{min}\}$ based on fundamental knowledge of advanced algebra. As $x$ changes in $\mathcal{X}$, this upper bound of rank will never be exceeded. Hence the rank of $H(x)$ is at most $\min\{l, d_{min}\}$, when $H(x)$ is considered as a matrix of functions on $x$. $\qquad\square$

## A.2 PROOF OF THEOREM 2

*Proof.* We first verify that the functional $\langle \cdot, \cdot \rangle$ defined in (14) satisfies the following three fundamental criteria of an inner product for any $H, G, F \in \mathcal{H}$ and any scalars $\mathfrak{a}, \mathfrak{b} \in \mathbb{R}$.

(1) Symmetry:

$$\langle H, G \rangle = \int_{\mathcal{X}} \mathrm{tr}(H(x)^\top G(x)) \, \mathrm{d}\nu(x) = \int_{\mathcal{X}} \mathrm{tr}(G(x)^\top H(x)) \, \mathrm{d}\nu(x) = \langle G, H \rangle. \qquad (25)$$

(2) Linearity in the first argument:

$$\begin{aligned}
&\langle \mathfrak{a}H + \mathfrak{b}F, G \rangle \\
&= \int_{\mathcal{X}} \mathrm{tr}((\mathfrak{a}H(x) + \mathfrak{b}F(x))^\top G(x)) \, \mathrm{d}\nu(x) \\
&= \int_{\mathcal{X}} \mathrm{tr}((\mathfrak{a}H(x))^\top G(x)) \, \mathrm{d}\nu(x) + \int_{\mathcal{X}} \mathrm{tr}((\mathfrak{b}F(x))^\top G(x)) \, \mathrm{d}\nu(x) \\
&= \mathfrak{a}\int_{\mathcal{X}} \mathrm{tr}(H(x)^\top G(x)) \, \mathrm{d}\nu(x) + \mathfrak{b}\int_{\mathcal{X}} \mathrm{tr}(F(x)^\top G(x)) \, \mathrm{d}\nu(x) \\
&= \mathfrak{a}\langle H, G \rangle + \mathfrak{b}\langle F, G \rangle.
\end{aligned} \qquad (26)$$

(3) Positive-definiteness: suppose $H \neq 0$, which means that the measure $\nu\{x \in \mathcal{X} : H(x) \neq 0\} > 0$ in the sense that $H \in \mathcal{H}$. It can be seen that $\{x \in \mathcal{X} : \mathrm{tr}(H(x)^\top H(x)) \neq 0\} = \{x \in \mathcal{X} : H(x) \neq 0\}$, thus $\nu\{x \in \mathcal{X} : \mathrm{tr}(H(x)^\top H(x)) \neq 0\} > 0$. Then

$$\langle H, H \rangle = \int_{\mathcal{X}} \mathrm{tr}(H(x)^\top H(x)) \, \mathrm{d}\nu(x) > 0. \qquad (27)$$

Hence $\langle H, H \rangle$ is positive definite w.r.t. $H$.

Second, we investigate the completeness of $\mathcal{H}$ under the induced norm $\|H\|_{\mathcal{H}}$. Direct calculation yields

$$\|H\|_{\mathcal{H}} = \sqrt{\int_{\mathcal{X}} \mathrm{tr}(H(x)^\top H(x)) \, \mathrm{d}\nu(x)} = \sqrt{\sum_{i,j} \int_{\mathcal{X}} H_{ij}^2(x) \, \mathrm{d}\nu(x)}. \qquad (28)$$

Since $H_{ij}$ belongs to the complete metric space $L^2(\mathcal{X})$, any Cauchy sequence $\{H_{(k),ij}\}_{k=1}^\infty$ converges to some $H_{(*),ij} \in L^2(\mathcal{X})$ under the $L^2$ norm

$$\|H_{ij}\|_2 := \sqrt{\int_{\mathcal{X}} H_{ij}^2(x) \, \mathrm{d}\nu(x)}. \qquad (29)$$

It can be seen that

$$\begin{aligned}
\|H_{(k)}\|_{\mathcal{H}} &\geqslant \|H_{(k),ij}\|_2, \ \forall i, j; \\
\|H_{(k)}\|_{\mathcal{H}} \to 0 \quad &\implies \quad \|H_{(k),ij}\|_2 \to 0, \ \forall i, j,
\end{aligned} \qquad (30)$$

which means that the norm $\|H\|_{\mathcal{H}}$ dominates the norm set $\{\|H_{ij}\|_2 : \forall i, j\}$.

Given any Cauchy sequence $\{H_{(k)}\}_{k=1}^\infty$, we have $\|H_{(m)} - H_{(n)}\|_{\mathcal{H}} \to 0$ as $m, n \to 0$. It follows from (30) that $\|H_{(m),ij} - H_{(n),ij}\|_2 \to 0$ as $m, n \to 0$ for all $i, j$. Applying the completeness of

$L^2(\mathcal{X})$ to the Cauchy sequence of each index $\{H_{(k),ij}\}_{k=1}^{\infty}$, we have $H_{(k),ij} \to H_{(*),ij} \in L^2(\mathcal{X})$. Collecting all the indices $i, j$, we have $H_{(k)} \to H_{(*)} \in \mathcal{H}$ under the norm $\|\cdot\|_{\mathcal{H}}$. Hence $\mathcal{H}$ is a complete metric space, i.e., a Hilbert space under the inner product $\langle \cdot, \cdot \rangle$ in (14).

$\square$

### A.3 PROOF OF THEOREM 3

*Proof.* From Theorem 1, we know that the rank of $H(z)$ is no larger than $\min\{l, d_{min}\}$ when $z$ varies in $\mathcal{B}_p[x; \epsilon]$. Hence there exists a largest rank $R \leqslant \min\{l, d_{min}\}$ for all $z \in \mathcal{B}_p[x; \epsilon]$. Given $H \in \mathcal{H}$, we consider $H(x)$ as a matrix and conduct an SVD of $H(x) := \sum_{r=1}^{R'} \sigma_r \psi_r$ for some $R' \leqslant R$. This $R'$ can be seen as the essential rank of $H(x)$. Note that if $R' < R$, we can let $\sigma_r = \sigma_{R'}$ and $\psi_r = [0]_{l \times d_{min}}$ for all $(R' + 1) \leqslant r \leqslant R$. Then $H(x) := \sum_{r=1}^{R} \sigma_r \psi_r$ for the unified rank $R$ with $\sigma_r > 0$ for all $r \in [R] := \{1, 2, \cdots, R\}$.

For any $z \in \mathcal{B}_p[x; \epsilon]$ and $z \neq x$, there is also an SVD of $H(z) := \sum_{r=1}^{R} \sigma'_r \psi'_r$. We can directly define a set of functions depending on $z$:

$$\Psi_r(z) := \frac{\sigma'_r}{\sigma_r} \cdot \psi'_r, \quad \forall r \in [R]. \tag{31}$$

As $z$ varies in $\mathcal{B}_p[x; \epsilon]$, $\Psi_r(z)$ is fit to the local SVD of $H(z)$ and extends the basis $\psi_r$ from one point $x$ to its neighborhood $\mathcal{B}_p[x; \epsilon]$. Hence (31) yields (15). Note that $\{\sigma_r\}_{r=1}^{R}$ and $\{\Psi_r(z)\}_{r=1}^{R}$ are dependent on $x$, since they are generated from $H(x) = \sum_{r=1}^{R} \sigma_r \psi_r$. By this means, the decomposition (15) enforces a set of constant singular values and embeds the varying components in the basis $\{\Psi_r(z)\}_{r=1}^{R}$.

Last, we need to verify that $\{\Psi_r\}_{r=1}^{R}$ are mutually orthogonal in the Hilbert space $\mathcal{H}$. As a fundamental property of SVD, $\{\Psi_r(z)\}_{r=1}^{R}$ are mutually orthogonal at any fixed $z \in \mathcal{B}_p[x; \epsilon]$ in the sense of matrix calculation:

$$\text{tr}(\Psi_r(z)^\top \Psi_r(z)) = \text{tr}(\Psi_r(z) \Psi_r(z)^\top) = 1,$$
$$\text{tr}(\Psi_r(z)^\top \Psi_{r'}(z)) = \text{tr}(\Psi_{r'}(z) \Psi_r(z)^\top) = 0, \quad \forall r, \ \forall r' \neq r, \ \forall z \in \mathcal{B}_p[x; \epsilon]. \tag{32}$$

Based on Theorem 2, we can directly calculate the following inner products:

$$\langle \Psi_r, \Psi_r \rangle = \int_{\mathcal{B}_p[x;\epsilon]} \text{tr}(\Psi_r(z)^\top \Psi_r(z)) \, d\nu(z) = \int_{\mathcal{B}_p[x;\epsilon]} 1 \cdot d\nu(z) = \nu(\mathcal{B}_p[x; \epsilon]) > 0,$$

$$\langle \Psi_r, \Psi_{r'} \rangle = \int_{\mathcal{B}_p[x;\epsilon]} \text{tr}(\Psi_r(z)^\top \Psi_{r'}(z)) \, d\nu(z) = \int_{\mathcal{B}_p[x;\epsilon]} 0 \cdot d\nu(z) = 0, \quad \forall r, \ \forall r' \neq r, \tag{33}$$

which indicates that $\{\Psi_r\}_{r=1}^{R}$ are mutually orthogonal in the Hilbert space $\mathcal{H}$.

$\square$

### A.4 PROOF OF THEOREM 5

*Proof.* **Part (1):** We first verify that $\nabla \mathfrak{f}(H) = H$. According to Definition 4,

$$\lim_{\substack{\bar{H} \xrightarrow{\mathcal{H}} H \\ \bar{H} \neq H}} \frac{\frac{1}{2}\|\bar{H}\|_{\mathcal{H}}^2 - \frac{1}{2}\|H\|_{\mathcal{H}}^2 - \langle H, \bar{H} - H \rangle}{\|\bar{H} - H\|_{\mathcal{H}}}$$

$$= \lim_{\substack{\bar{H} \xrightarrow{\mathcal{H}} H \\ \bar{H} \neq H}} \frac{\frac{1}{2}(\|H\|_{\mathcal{H}}^2 + 2\langle H, \bar{H} - H \rangle + \|\bar{H} - H\|_{\mathcal{H}}^2) - \frac{1}{2}\|H\|_{\mathcal{H}}^2 - \langle H, \bar{H} - H \rangle}{\|\bar{H} - H\|_{\mathcal{H}}}$$

$$= \lim_{\substack{\bar{H} \xrightarrow{\mathcal{H}} H \\ \bar{H} \neq H}} \frac{\frac{1}{2}\|\bar{H} - H\|_{\mathcal{H}}^2}{\|\bar{H} - H\|_{\mathcal{H}}}$$

$$= \lim_{\substack{\bar{H} \xrightarrow{\mathcal{H}} H \\ \bar{H} \neq H}} \frac{1}{2} \|\bar{H} - H\|_{\mathcal{H}} = 0. \tag{34}$$

Hence $\nabla\mathfrak{f}(H) = H$. Moreover, the above deduction also implies that

$$\frac{1}{2}\|\bar{H}\|_{\mathcal{H}}^2 - \frac{1}{2}\|H\|_{\mathcal{H}}^2 - \langle H, \bar{H} - H \rangle = \frac{1}{2}\|\bar{H} - H\|_{\mathcal{H}}^2 \geqslant 0, \quad \forall H, \bar{H} \in \mathcal{H}. \tag{35}$$

It follows from the first-order condition of convexity that $\mathfrak{f}(H)$ is convex w.r.t. $H$.

**Part (2):** The functional $\mathfrak{g}(H) = \sum_{r=1}^{R'} \sigma_r(x) = \|H(x)\|_*$ is actually the nuclear norm of the matrix $H(x)$ (see Appendix A.3). Then for any $H, F \in \mathcal{H}$ and any $\gamma \in (0, 1)$,

$$\begin{aligned}
&\mathfrak{g}(\gamma H + (1 - \gamma)F) \\
=& \|\gamma H(x) + (1 - \gamma)F(x)\|_* \\
\leqslant& \gamma\|H(x)\|_* + (1 - \gamma)\|F(x)\|_* \\
=& \gamma\mathfrak{g}(H) + (1 - \gamma)\mathfrak{g}(F),
\end{aligned} \tag{36}$$

which indicates that $\mathfrak{g}(\cdot)$ is convex.

$\square$

### A.5 Proof of Theorem 6

*Proof.* Appendix A.4 indicates that $\mathfrak{g}(G) = \|G(x)\|_*$ is actually the nuclear norm of a matrix, whose subdifferential has been well studied (Watson, 1992; Cai et al., 2010):

$$\partial\|G(x)\|_* = \{UV^\top + Z : Z \in \mathbb{R}^{l \times d_{min}}, U^\top Z = 0, ZV = 0, \|Z\|_2 \leqslant 1\}, \tag{37}$$

where the columns of $U$ and $V$ represent the left and right singular vectors of $G(x)$, respectively. Theorem 5 indicates that the objective function in (17) is convex w.r.t. $G$. Then from Fermat's rule, $G$ is a solution to (17) if and only if it satisfies the following relationship regarding the subdifferential $\partial\mathfrak{g}(G)$:

$$\begin{aligned}
(G - H) + \tau\partial\mathfrak{g}(G) &\ni 0, \\
H - G &\in \tau\partial\mathfrak{g}(G).
\end{aligned} \tag{38}$$

Since $\mathfrak{g}$ is a functional supported on a single-point $x$, (38) can be further reduced to the tangent space at $x$:

$$H(x) - G(x) \in \tau\partial\|G(x)\|_*. \tag{39}$$

From Theorem 3, $H$ can be decomposed into two parts:

$$\begin{aligned}
H(z) &:= \sum_{r=1}^{\bar{R}} \sigma_r \Psi_r(z) + \sum_{r=\bar{R}+1}^{R} \sigma_r \Psi_r(z), \\
\sigma_r &\begin{cases} > \tau, & 1 \leqslant r \leqslant \bar{R}, \\ \leqslant \tau, & \bar{R} + 1 \leqslant r \leqslant R, \end{cases}
\end{aligned} \tag{40}$$

where the first part corresponds to the singular values that are larger than $\tau$, while the second part corresponds to the singular values that are smaller than or equal to $\tau$. Then we construct the following term:

$$\hat{G}(z) := \sum_{r=1}^{\bar{R}} (\sigma_r - \tau)\Psi_r(z). \tag{41}$$

We have

$$H(z) - \hat{G}(z) = \tau\left(\sum_{r=1}^{\bar{R}} \Psi_r(z) + \sum_{r=\bar{R}+1}^{R} \frac{\sigma_r}{\tau}\Psi_r(z)\right) =: \tau(H_{[1]}(z) + H_{[2]}(z)). \tag{42}$$

By letting $z = x$ in (42), $H_{[1]}(x)$ and $H_{[2]}(x)$ correspond to the $UV^\top$ and $Z$ in (37). First, the SVD of matrix $H(x)$ yields

$$\sum_{r=1}^{\bar{R}} \Psi_r(x) = \sum_{r=1}^{\bar{R}} u_r v_r^\top = UV^\top. \tag{43}$$

Second, since $\{\Psi_r\}_{r=1}^{\bar{R}}$ and $\{\Psi_r\}_{r=\bar{R}+1}^{R}$ are mutually orthogonal, we have

$$U^\top H_{[2]}(x) = 0, H_{[2]}(x)V = 0. \tag{44}$$

Third, it is obvious that all the singular values of $H_{[2]}(x)$ are no larger than 1. In summary, $\hat{G}$ defined in (41) satisfies (39), thus it is a solution to (17). Note that the above deduction already considers the situation that the essential rank $R'$ of $H(x)$ may be smaller than $R$, because the corresponding $\{\Psi_r(x)\}_{r=R'+1}^{R}$ are set to 0 and $\sum_{r=R'+1}^{R} \sigma_r \Psi_r(x)$ will vanish.

$\square$

### A.6 PROOF OF THEOREM 7

*Proof.* **Part (1):** We illustrate how the algorithm (20) is developed and prove that it converges weakly to a solution to (19). We decompose the iteration (20) as a composite of two operators:

$$\mathcal{T}_1(G) := G + \varsigma\mathrm{proj}_\Omega(H - G), \tag{45}$$

$$\mathcal{T}_2(G) := \mathrm{prox}_{\varsigma\tau\mathfrak{g}(\cdot)}(G), \tag{46}$$

$$G_{(k+1)} = \mathcal{T}(G_{(k)}) := \mathcal{T}_2 \circ \mathcal{T}_1(G_{(k)}). \tag{47}$$

We need to verify that a solution to (19) is also a fixed point of $\mathcal{T}$:

$$\mathrm{Fix}\mathcal{T} := \{G \in \mathcal{H} : G = \mathcal{T}(G)\}. \tag{48}$$

Condition 1 implies that $\mathfrak{g}(G)$ is a continuous function on $G \in \mathcal{H}$, based on (14), (15), and Theorem 5 (2). We define the following functional:

$$\mathfrak{f}_\Omega(G) := \frac{1}{2}\|\mathrm{proj}_\Omega(G - H)\|_\mathcal{H}^2. \tag{49}$$

Applying Theorem 5 (1) to the index set $\Omega$, we have $\nabla\mathfrak{f}_\Omega(G) = \mathrm{proj}_\Omega(G - H)$. Then it can be seen that the objective function $\mathfrak{h}(G)$ in (19) is continuous, convex, nonnegative and coercive (i.e., $\mathfrak{h}(G) \to +\infty$ as $\|G\|_\mathcal{H} \to +\infty$), thus there exists at least one solution $G^*$ to (19). Then from Fermat's rule,

$$0 \in \partial\left\{\frac{1}{2}\|\mathrm{proj}_\Omega(G^* - H)\|_\mathcal{H}^2 + \tau\mathfrak{g}(G^*)\right\}$$

$$= \nabla\left\{\frac{1}{2}\|\mathrm{proj}_\Omega(G^* - H)\|_\mathcal{H}^2\right\} + \tau\partial\mathfrak{g}(G^*) \tag{50}$$

$$= \mathrm{proj}_\Omega(G^* - H) + \tau\partial\mathfrak{g}(G^*). \tag{51}$$

The equality in (50) is from a basic property of subdifferential that $\partial(\mathfrak{f} + \mathfrak{g}) = \nabla\mathfrak{f} + \partial\mathfrak{g}$ (Exercise 8.8 (c) of Rockafellar & Wets 2009). We further transform (51) towards (20) by introducing the learning rate $\varsigma \in (0, 1)$:

$$\mathrm{proj}_\Omega(H - G^*) \in \tau\partial\mathfrak{g}(G^*)$$

$$\Leftrightarrow \quad \varsigma\mathrm{proj}_\Omega(H - G^*) \in \varsigma\tau\partial\mathfrak{g}(G^*)$$

$$\Leftrightarrow \quad (G^* + \varsigma\mathrm{proj}_\Omega(H - G^*)) - G^* \in \varsigma\tau\partial\mathfrak{g}(G^*). \tag{52}$$

Since $\varsigma\tau\mathfrak{g}(\cdot)$ is convex, (52) is equivalent to the following:

$$G^* = \mathrm{prox}_{\varsigma\tau\mathfrak{g}(\cdot)}(G^* + \varsigma\mathrm{proj}_\Omega(H - G^*)), \tag{53}$$

which is from a basic property of proximal mapping that

$$F \in \partial\mathfrak{g}(G) \Leftrightarrow G = \mathrm{prox}_{\mathfrak{g}(\cdot)}(G + F). \tag{54}$$

Hence $G^* \in \text{Fix}\mathcal{T}$. Conversely, for any $G^* \in \text{Fix}\mathcal{T}$, it is also a solution to (19) based on the above deduction. To summarize, $\text{Fix}\mathcal{T}$ is the nonempty solution set of (19). The algorithm (20) is developed based on this fixed-point property.

To prove the convergence of (19), we further decompose $\mathcal{T}_1$ in (45) as follows:

$$\begin{aligned}
\mathcal{T}_1(G) &= (1-\varsigma)G + (\varsigma G - \varsigma\text{proj}_\Omega(G)) + \varsigma\text{proj}_\Omega(H) \\
&= (1-\varsigma)G + \varsigma\text{proj}_{\bar{\Omega}}(G) + \varsigma\text{proj}_\Omega(H) \\
&= (1-\varsigma)G + \varsigma(\text{proj}_{\bar{\Omega}}(G) + \text{proj}_\Omega(H)),
\end{aligned} \tag{55}$$

where $\bar{\Omega}$ denotes the complementary index set of $\Omega$. We define a new operator

$$\mathcal{T}_3(G) := \text{proj}_{\bar{\Omega}}(G) + \text{proj}_\Omega(H). \tag{56}$$

Then

$$\|\mathcal{T}_3(G) - \mathcal{T}_3(F)\|_\mathcal{H}^2 = \|\text{proj}_{\bar{\Omega}}(G) - \text{proj}_{\bar{\Omega}}(F)\|_\mathcal{H}^2 \leqslant \|G - F\|_\mathcal{H}^2, \quad \forall G, F \in \mathcal{H}. \tag{57}$$

The inequality holds from the definition in (14). According to the knowledge of non-expansive operators (Bauschke & Combettes, 2017), $\mathcal{T}_3$ is a non-expansive operator. Then the decomposition (55) becomes

$$\mathcal{T}_1 = (1-\varsigma)\mathcal{I} + \varsigma\mathcal{T}_3, \tag{58}$$

where $\mathcal{I}$ is the identity operator. Such an operator as $\mathcal{T}_1$ is called a $\varsigma$-averaged non-expansive operator. Besides, $\mathcal{T}_2$ in (46) is a proximal mapping w.r.t. a convex function $\varsigma\tau\mathfrak{g}(\cdot)$, which is a $\frac{1}{2}$-averaged non-expansive operator. Thus the composite $\mathcal{T} = \mathcal{T}_2 \circ \mathcal{T}_1$ in (47) is also a $\frac{1}{2-s}$-averaged non-expansive operator. According to the Krasnosel'skiĭ-Mann's theorem (Mann, 1953; Krasnosel'skiĭ, 1955), the sequence $\{G_{(k)}\}_{k=1}^\infty$ generated by the averaged non-expansive operator $\mathcal{T}$ converges weakly to a point in $\text{Fix}\mathcal{T}$, which is a solution to (19).

**Part (2):** We investigate the descent property of the algorithm (20). Direct calculation yields

$$\begin{aligned}
&\|\nabla\mathfrak{f}_\Omega(G) - \nabla\mathfrak{f}_\Omega(F)\|_\mathcal{H}^2 \\
&= \|\text{proj}_\Omega(G - H) - \text{proj}_\Omega(F - H)\|_\mathcal{H}^2 \\
&= \|\text{proj}_\Omega(G - F)\|_\mathcal{H}^2 \\
&\leqslant \|G - F\|_\mathcal{H}^2, \quad \forall G, F \in \mathcal{H}.
\end{aligned} \tag{59}$$

Hence $\nabla\mathfrak{f}_\Omega$ is Lipschitz continuous with the Lipschitz constant $\mathscr{L}_\mathfrak{f} = 1$. Then from Proposition A.24 of (Bertsekas, 1999),

$$\mathfrak{f}_\Omega(F) \leqslant \mathfrak{f}_\Omega(G) + \langle\nabla\mathfrak{f}_\Omega(G), F - G\rangle + \frac{1}{2}\|G - F\|_\mathcal{H}^2, \quad \forall G, F \in \mathcal{H}. \tag{60}$$

Inserting $F = G_{(k+1)}$ and $G = G_{(k)}$ into (60), we have

$$\mathfrak{f}_\Omega(G_{(k+1)}) \leqslant \mathfrak{f}_\Omega(G_{(k)}) + \langle\nabla\mathfrak{f}_\Omega(G_{(k)}), G_{(k+1)} - G_{(k)}\rangle + \frac{1}{2}\|G_{(k)} - G_{(k+1)}\|_\mathcal{H}^2. \tag{61}$$

On the other hand, (20) can be reformulated as follows according to the definition of proximal mapping (17):

$$\begin{aligned}
&\frac{1}{2}\|G_{(k+1)} - (G_{(k)} + \varsigma\text{proj}_\Omega(H - G_{(k)}))\|_\mathcal{H}^2 + \varsigma\tau\mathfrak{g}(G_{(k+1)}) \\
&\leqslant \frac{1}{2}\|G_{(k)} - (G_{(k)} + \varsigma\text{proj}_\Omega(H - G_{(k)}))\|_\mathcal{H}^2 + \varsigma\tau\mathfrak{g}(G_{(k)}) \\
\Leftrightarrow \quad &\langle G_{(k+1)} - G_{(k)}, \varsigma\text{proj}_\Omega(H - G_{(k)})\rangle + \frac{1}{2}\|G_{(k+1)} - G_{(k)}\|_\mathcal{H}^2 + \varsigma\tau\mathfrak{g}(G_{(k+1)}) \\
&\leqslant \varsigma\tau\mathfrak{g}(G_{(k)}) \\
\Leftrightarrow \quad &\langle G_{(k+1)} - G_{(k)}, \text{proj}_\Omega(H - G_{(k)})\rangle + \frac{1}{2\varsigma}\|G_{(k+1)} - G_{(k)}\|_\mathcal{H}^2 + \tau\mathfrak{g}(G_{(k+1)}) \\
&\leqslant \tau\mathfrak{g}(G_{(k)}).
\end{aligned} \tag{62}$$

Summing up both sides of (61) and (62), we have

$$\mathfrak{h}(G_{(k+1)}) \leqslant \mathfrak{h}(G_{(k)}) + (\frac{1}{2} - \frac{1}{2\varsigma})\|G_{(k+1)} - G_{(k)}\|_{\mathcal{H}}^2. \tag{63}$$

Since $\varsigma < 1$, we have $\mathfrak{h}(G_{(k+1)}) \leqslant \mathfrak{h}(G_{(k)})$, which is the descent property of the algorithm.

**Part (3):** We derive the convergence rate for the objective value. Recall $G^*$ as a solution to (19). Since $\mathfrak{f}_\Omega$ is convex, we have

$$\mathfrak{f}_\Omega(G_{(k)}) - \mathfrak{f}_\Omega(G^*) + \langle G^* - G_{(k)}, \nabla\mathfrak{f}_\Omega(G_{(k)})\rangle \leqslant 0. \tag{64}$$

Summing up both sides of (61) and (64), we have

$$\mathfrak{f}_\Omega(G_{(k+1)}) - \mathfrak{f}_\Omega(G^*) + \langle G^* - G_{(k+1)}, \nabla\mathfrak{f}_\Omega(G_{(k)})\rangle \leqslant \frac{1}{2}\|G_{(k)} - G_{(k+1)}\|_{\mathcal{H}}^2$$
$$\leqslant \frac{1}{2\varsigma}\|G_{(k)} - G_{(k+1)}\|_{\mathcal{H}}^2. \tag{65}$$

By using (54), (20) can be reformulated as follows:

$$G_{(k+1)} = \text{prox}_{\varsigma\tau\mathfrak{g}(\cdot)}(G_{(k)} + \varsigma\text{proj}_\Omega(H - G_{(k)}))$$
$$\Leftrightarrow \quad G_{(k)} + \varsigma\text{proj}_\Omega(H - G_{(k)}) - G_{(k+1)} \in \varsigma\tau\mathfrak{g}(G_{(k+1)})$$
$$\Leftrightarrow \quad \frac{1}{\varsigma}(G_{(k)} - G_{(k+1)}) + \text{proj}_\Omega(H - G_{(k)}) \in \tau\mathfrak{g}(G_{(k+1)}). \tag{66}$$

By the convexity of $\tau\mathfrak{g}(\cdot)$ and the property of subdifferential for a convex function,

$$\langle\frac{1}{\varsigma}(G_{(k)} - G_{(k+1)}) + \text{proj}_\Omega(H - G_{(k)}), G^* - G_{(k+1)}\rangle \leqslant \tau\mathfrak{g}(G^*) - \tau\mathfrak{g}(G_{(k+1)}). \tag{67}$$

Summing up both sides of (65) and (67), we have

$$\mathfrak{h}(G_{(k+1)}) - \mathfrak{h}(G^*) \leqslant \frac{1}{2\varsigma}\|G_{(k)} - G_{(k+1)}\|_{\mathcal{H}}^2 - \frac{1}{\varsigma}\langle G_{(k)} - G_{(k+1)}, G^* - G_{(k+1)}\rangle$$
$$= \frac{1}{2\varsigma}(\|G_{(k)} - G^*\|_{\mathcal{H}}^2 - \|G^* - G_{(k+1)}\|_{\mathcal{H}}^2). \tag{68}$$

Summing up both sides of (68) from $k = 1$ to $k = K$, we have

$$\sum_{k=1}^{K}(\mathfrak{h}(G_{(k+1)}) - \mathfrak{h}(G^*)) \leqslant \frac{1}{2\varsigma}(\|G_{(1)} - G^*\|_{\mathcal{H}}^2 - \|G^* - G_{(K+1)}\|_{\mathcal{H}}^2)$$
$$\leqslant \frac{1}{2\varsigma}\|G_{(1)} - G^*\|_{\mathcal{H}}^2. \tag{69}$$

On the other hand, we have verified that $\{\mathfrak{h}(G_{(k)})\}_{k=1}^{\infty}$ is a non-increasing sequence in Part (2), thus

$$\sum_{k=1}^{K}(\mathfrak{h}(G_{(k+1)}) - \mathfrak{h}(G^*)) \geqslant K(\mathfrak{h}(G_{(K+1)}) - \mathfrak{h}(G^*)). \tag{70}$$

Combining (69) and (70), we have

$$\mathfrak{h}(G_{(K+1)}) - \mathfrak{h}(G^*) \leqslant \frac{1}{2\varsigma K}\|G_{(1)} - G^*\|_{\mathcal{H}}^2. \tag{71}$$

Hence $\{\mathfrak{h}(G_{(k)})\}_{k=1}^{\infty}$ converges to the optimal objective value $\mathfrak{h}(G^*)$ with a sublinear rate $\mathcal{O}(\frac{1}{k})$.

$\square$

## A.7 EXPERIMENTAL SETTINGS

In all the experiments, both the spectral regularization strength $\tau$ and the learning rate $\varsigma$ in (20) are empirically set to 0.1. Table A1 shows that the performance of FSIO is stable to a small change of $\tau$. The hyperparameter $\lambda_0$ in (21) is empirically set to $10^5$. The full index set is used as $\Omega$ for FSIO.

Table A1: Attack success rate of FSIO with respect to the spectral regularization strength $\tau$ on MNIST (ReLU).

| Attack Success Rate (%) | 96.21 | 97.50 | **98.18** | 97.80 | 96.89 |
|---|---|---|---|---|---|
| $\tau$ | 0.08 | 0.09 | **0.1** | 0.11 | 0.12 |

Other hyperparameters like $\lambda_1$, $\alpha_x$, $\alpha_{H'}$, $\alpha_{\lambda_1}$ are all set to the same as those of (Yu & Gao, 2024). In each iteration of training, the proposed FSIO is used to extract the principal component of the activation map and calculate the loss function (22) at first, and then the adversarial training scheme of (Yu & Gao, 2024) is used to construct perturbed inputs.

In the adversarial attack experiments on image classification (Section 4.1), we consider both untargeted and targeted attacks. The former misleads the classifier to any of the incorrect classes, while the latter misleads the classifier to a specific class. The pure objective $J$ is set as the cross-entropy loss w.r.t. the true labels for untargeted attacks, or the negation of cross-entropy loss w.r.t. the targeted labels for targeted attacks.

At each attack iteration, FSIO requires a computational complexity of $\mathcal{O}(\min\{l, d_{min}\} \cdot l \cdot d_{min} + \sum_{i=1}^{l}(d_{i-1} \cdot d_i))$. The first term corresponds to the SVD specially for FSIO, while the second term corresponds to the activation value computation and is the same for other compared methods. However, $\min\{l, d_{min}\} \cdot l \cdot d_{min} \leqslant l \cdot d_{min}^2 \leqslant \sum_{i=1}^{l}(d_{i-1} \cdot d_i)$, thus FSIO actually has the same order of complexity $\sum_{i=1}^{l}(d_{i-1} \cdot d_i)$ as that of other compared methods.

## A.8 ABLATION STUDY

Ablation experiments are conducted on randomized NNs that produce scalar outcomes to demonstrate the importance of each part in FSIO. Specifically, this task is to maximize the scalar output of a randomized NN $F : \mathbb{R}^n \mapsto \mathbb{R}$ with network parameters uniformly sampled from the range $(-1, 1)$ and input $x$ constrained in the interval $[-1, 1]^n$. Two ablated versions of FSIO are considered: **M1** - detaching the subgradient of $L^*$ w.r.t. $H'$, and **M2** - detaching the gradient of $L^*$ w.r.t. $x$. Besides, ADR-GD can be seen as the ablated version of FSIO by removing the entire functional shrinkage module. Results in Table A2 show that FSIO outperforms other common optimizers, competitors, and ablated versions in all three models. For example, the performance of M1 or M2 deteriorates significantly where the (sub)gradient w.r.t. $H'$ or $x$ is detached. Besides, the gap between FSIO and ADR-GD also indicates that the proposed functional shrinkage module is effective in input optimization.

Table A2: Optimized objective values of different methods and ablated versions on three randomized neural networks.

| Model A | $[[10, 64], [64, 64], [64, 1]]$ |
|---|---|
| Model B | $[[10, 500], [500, 500], [500, 500], [500, 1]]$ |
| Model C | $[[128, 500], [500, 500], [500, 500], [500, 1]]$ |

| | Model A | Model B | Model C |
|---|---|---|---|
| GD | 17.91 | 1578.94 | 34210.52 |
| Adam | 17.86 | 1587.05 | 37277.69 |
| Adagrad | 14.46 | 1616.11 | 37752.65 |
| Perturbed GD (Jin et al., 2017) | 14.41 | 1561.16 | 35533.07 |
| ADR-GD | 18.01 | 1601.22 | 40531.88 |
| M1 | 14.66 | 177.94 | 11416.01 |
| M2 | -8.48 | -62.88 | -82.96 |
| **FSIO (ours)** | **21.83** | **2614.83** | **44295.57** |

## A.9 ADDITIONAL EXPERIMENTS

We further add two experimental scenarios to evaluate the extendability of FSIO, shown in Table A3. The first scenario is to attack the Deep Q-Network (DQN, Mnih et al. 2013) in the Freeway Atari-game (Bellemare et al., 2013). The perturbation budget for all the relavant methods is set to $\epsilon = 3/255$. Results show that FSIO is effective in DQN, which typically handles discrete action spaces. The second scenario is to improve DRL of the PPO algorithm in the Walker2d environment, which corresponds to a continuous action space. Results show that FSIO is effective in PPO for the

continuous action task, which is consistent with the PPO scenarios for the discrete action spaces in Section 4.2.

Table A3: Average episode returns of different methods on attacking neural policies (upper) and improving deep reinforcement learning performance (lower).

| | | Clean | FGSM | PGD | APGD | AutoAttack | C&W | ADR-GD | **FSIO (ours)** |
|---|---|---|---|---|---|---|---|---|---|
| Freeway (DQN)↓ | MEAN | 21.80 | 14.77 | 13.52 | 13.35 | 13.38 | 10.98 | 12.29 | **9.55** |
| | STD | 1.75 | 1.19 | 1.09 | 1.08 | 1.07 | 0.88 | 0.99 | 0.76 |
| Walker2d-v4 (PPO)↑ | MEAN | 2847.20 | 2912.76 | 3115.88 | 3279.99 | 3499.32 | 3889.94 | 4494.95 | **6782.85** |
| | STD | 39.68 | 40.81 | 43.66 | 45.96 | 49.03 | 54.50 | 62.98 | 95.03 |

## A.10 VISUALIZATION OF CLEAN AND PERTURBED IMAGES

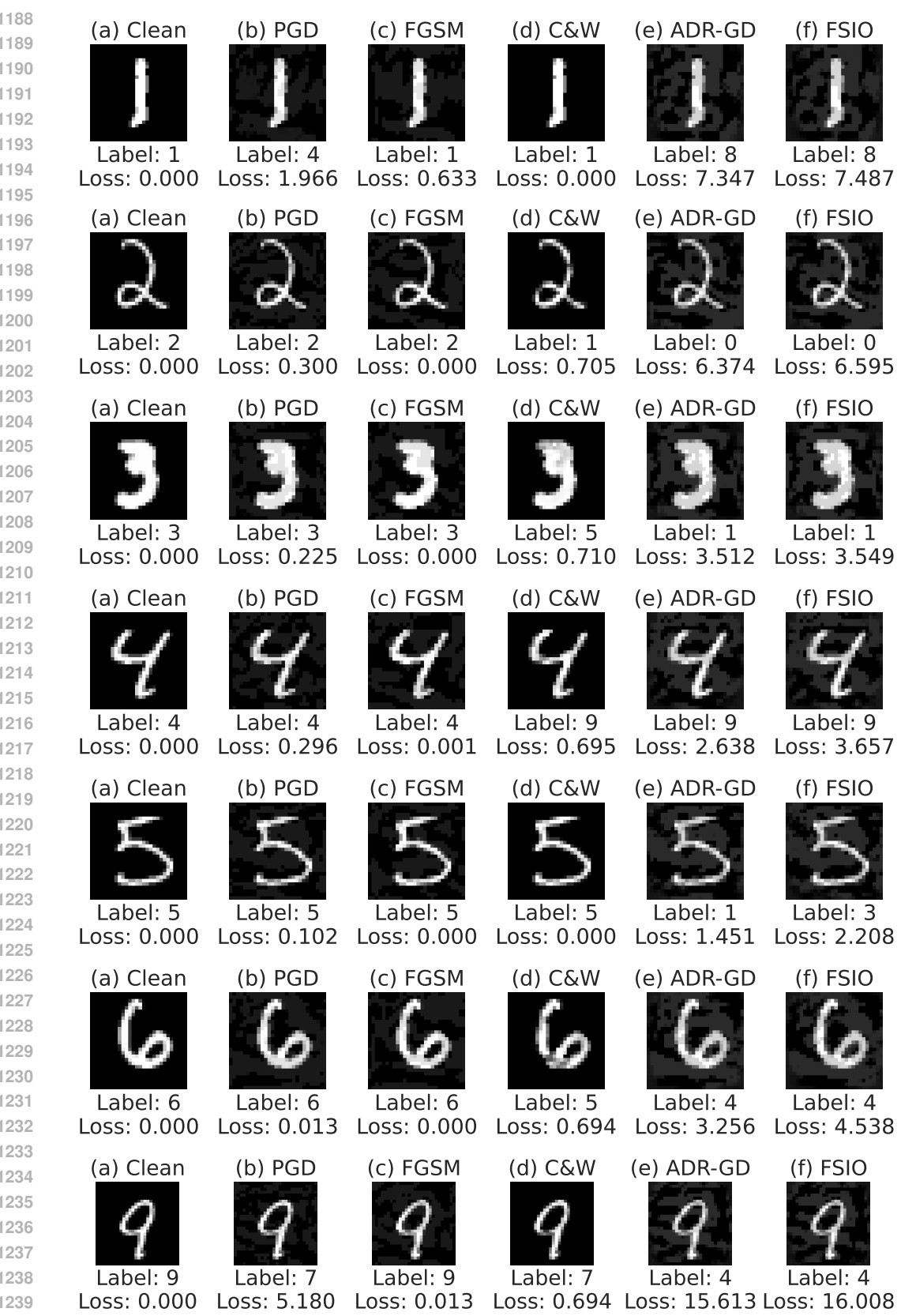

Figure A1: Visualization of clean and perturbed images constructed by different attack methods for MNIST with $\epsilon = 0.1$.

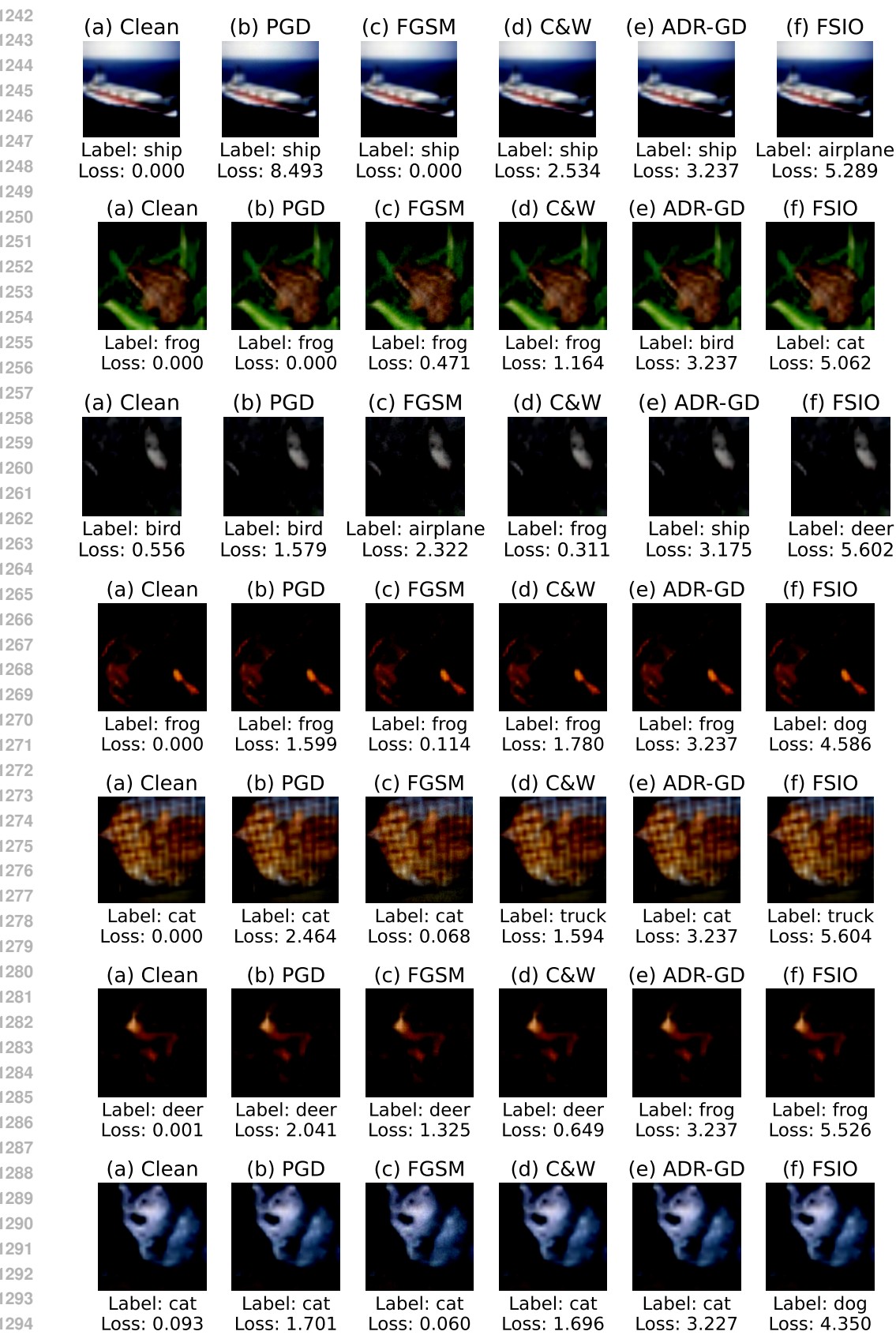

Figure A2: Visualization of clean and perturbed images constructed by different attack methods for CIFAR10 with $\epsilon = 8/255$.

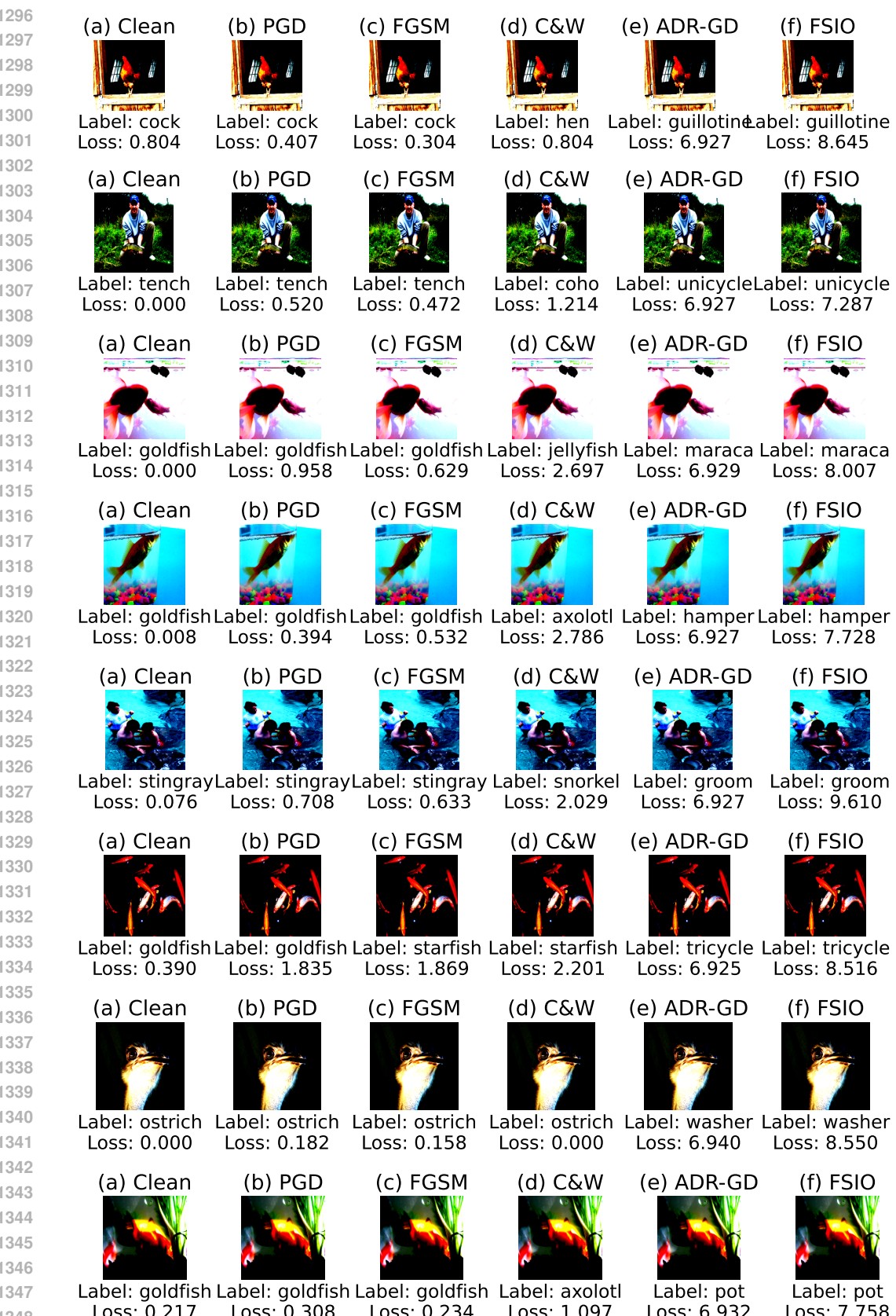

Figure A3: Visualization of clean and perturbed images constructed by different attack methods for ImageNet with $\epsilon = 4/255$.

