# OpenReview forum: "Functional Shrinkage for Input Optimization"
_ICLR.cc/2026/Conference — ICLR 2026 Conference Desk Rejected Submission_

### Official Review · Reviewer_i7HY · 2025-10-29

**Soundness:** 3
**Presentation:** 3
**Contribution:** 3
**Rating:** 6
**Confidence:** 3

**Summary:**

The manuscript proposes a functional shrinkage method (FSIO) that decomposes a neural network into a principal component containing the main structural information and dominant signals, and a residual component that includes outlier-sensitive information and noise, enabling adversarial attacks and input optimization. The method is proven to be sound through theoretical analysis and theorems. Comparative experimental results demonstrate that FSIO outperforms other methods with smaller perturbation budgets and is effective in capturing the vulnerable parts of neural networks, injecting perturbations into these parts. This provides new insight into the structural decomposition and weakness exploitation of neural networks.

**Strengths:**

1. The manuscript proposes a functional shrinkage method to extract the principal component of the network and inject perturbations into the residual component, providing a new insight into the structural decomposition and weaknesses of neural networks. Compared to other methods, it demonstrates superior performance.

2. The paper is easy to follow, with clear logical analysis and comprehensive theorem proofs. The consistency of the theoretical results is supported by comparative experimental validation.

3. The convergence proof and optimization model provides a guarantee for the soundness of the manuscript.

**Weaknesses:**

1. Based on the experimental results, the choice of activation function and the decomposition of activation patterns seem to have a significant impact on the method presented in the manuscript. Although the authors have considered activation functions such as ReLU, sigmoid, and softplus, FSIO's performance may not be optimal for certain types of activation functions.

2. FSIO injects perturbations into the residual component; however, this residual component may not fully cover all critical regions, which could result in missing some key attack targets.

**Questions:**

1. Could the shrinkage function be replaced by another form that retains these properties?

2. In the comparative experiments on "adversarial attacks on image classification," "adversarial attacks on neural policies," and "input optimization for deep reinforcement learning," the functional shrinkage method (FSIO) demonstrates strong performance. Could the authors analyze under what conditions the proposed method might fail or exhibit suboptimal performance?

3. In the comparative experiments for input optimization in deep reinforcement learning, why were DDPG and TD3 chosen as the algorithms for the reinforcement learning part? How would FSIO perform if applied to input optimization using PPO or DQN algorithms instead?

---

> ### Author Response · Authors · 2025-11-21
>
> We sincerely thank the reviewer for the careful reading and constructive feedback on our manuscript. We are pleased that the reviewer recognizes the novelty of the proposed FSIO method in decomposing neural networks (NNs) for adversarial attacks and input optimization, the soundness of the theoretical analysis, the clarity of the presentation, and the superior performance shown in the comparative experiments.
>
> **Q1.** While the root concept and foundation should adhere to the proposed framework and the theoretical properties, the proposed approach and idea might be extended to other mathematical functions (e.g., different types of soft-thresholding or non-linear filtering operations), which is a valuable direction for future research.
>
> **Q2.** For networks that are already intrinsically highly robust to all kinds of noise and perturbations, there would be little need to optimize the input, and FSIO might show little improvement. For an extreme but intuitive example, suppose there is a binary classification network that classifies any input as the positive class. Then no matter how FSIO optimizes the input, the network will just classify it as positive, thus there will be no improvement in this situation. However, there are considerable vulnerable parts in the NNs we investigate in the experiments, thus FSIO shows strong performance.
>
> **Q3.** We basically follow the standard evaluating criterion for input optimization ref. (Yu \& Gao, 2024) to be consistent, which uses DDPG and TD3 as algorithms. The main reason for using these two algorithms is that they are both based on the actor-critic framework, which features an explicit policy network (actor) that directly maps state to action.
>
> We also add experiments on the Deep Q-Network (DQN) and the Proximal Policy Optimization (PPO) algorithm in Appendix A.9 and Table A3. Since the standard DQN typically handles discrete action spaces, it cannot be directly used in the environments with continuous action spaces of Section 4.3. Instead, we use the Freeway Atari-game ref. (Bellemare et al., 2013) with a discrete action space for DQN. Results show that FSIO is effective in attacking DQN. As for PPO, it has already been tested in Section 4.2 for the discrete action spaces. Hence we further test it in the Walker2d environment with a continuous action space. Results show that FSIO is effective in PPO for both continuous and discrete action spaces.
>
> **Replies to Other Concerns in Weaknesses.** While the current work covers common activations (ReLU, Sigmoid, Softplus), we acknowledge that there are other types of activations that have not yet been examined due to the page limit and could be explored in the future. By definition, the "critical regions" missed by the residual component would be part of the network's dominant functional structure (the principal component). Introducing perturbations into these dominant regions would require a much larger perturbation budget to overcome the structural stability and change the output, which contradicts the goal of generating effective attacks with smaller budgets. Hence the success of FSIO lies in the residual component where a small perturbation could have the largest impact. From the extensive experimental results, this is the case across various machine learning tasks.
>
> Current experimental results span three different subfields, $22$ specific tasks, and six representative baselines, which covers sufficient diversity to prove the general utility of FSIO.

---

### Official Review · Reviewer_r3KP · 2025-10-29

**Soundness:** 2
**Presentation:** 2
**Contribution:** 2
**Rating:** 2
**Confidence:** 4

**Summary:**

This paper proposes FSIO (Functional Shrinkage for Input Optimization), a framework that formulates a neural network as a functional of its activation functions and applies a form of functional shrinkage to decompose it into principal and residual components. The authors claim that this decomposition enables more effective perturbation injection for input optimization tasks. The paper evaluates FSIO across three different domains: adversarial attacks on image classification, adversarial attacks on neural policies, and input optimization for deep reinforcement learning.

**Strengths:**

- The paper presents an ambitious and comprehensive framework that aims to unify multiple areas (e.g., adversarial attacks, neural verification, and reinforcement learning) under a single umbrella of input optimization. Typically, these approaches receive much recognition as they can lead to “cross-fertilization” between distinct areas.

- The theoretical formulation is mathematically rich, with detailed proofs and derivations for the proposed functional shrinkage process.

- The experimental evaluation covers a diverse set of application domains (vision and RL) and activation types (ReLU, Softplus, Sigmoid), which shows the authors’ effort to generalize their method beyond a single context.

**Weaknesses:**

**Lack of clarity and direction.**
A major weakness of the paper lies in its presentation and clarity. The manuscript devotes substantial space to mathematical formalism and proofs, but it does not provide an intuitive explanation of what the proposed method aims to achieve, why it matters, or how it connects to real-world optimization or robustness problems. The introduction and methodology sections are dense and abstract, with limited motivation for key design choices. As a result, it is difficult for a reader, even one familiar with adversarial machine learning, to grasp the central intuition behind FSIO and its intended contributions.
The paper indeed gives strong emphasis on formal derivations but provides little conceptual grounding. It introduces several functional operators and assumptions without explaining the rationale or the high-level mechanism that makes their FSIO approach beneficial for input optimization. In essence, the proofs dominate the exposition, while the link between these theoretical elements and the final algorithmic or empirical performance is underdeveloped. Consequently, it remains unclear why this approach should outperform existing optimization schemes or how it connects to known principles in, for example, adversarial robustness (as it is one of the tasks implemented).

**Weak experimental setup.**  The experimental section employs outdated or weak adversarial baselines [AttackBench]. Attacks such as FGSM and C&W are used, but modern strong baselines like APGD or AutoAttack are absent. Moreover, it is not clear how many iterations of PGD are being run, as a few iterations are insufficient for convergence and can bias results toward faster but weaker optimization schemes. The experiments also do not consider adversarially trained or robust models, which are now standard in attack evaluations. As a result, it is unclear whether the proposed FSIO method truly provides improvements beyond exploiting non-robust models or suboptimal attack configurations.

AttackBench Evaluating Gradient-based Attacks for Adversarial Examples, AAAI 2025.



**Lack of background and contextualization** The paper lacks sufficient technical and conceptual background for readers who are not already experts across all three subfields (adversarial robustness, DRL, and neural verification). It treats input optimization as a general umbrella that wishes to capture all of them, but does not clearly define how these tasks relate or why they should share a unified framework. The related work section also lacks a systematic discussion of prior works going in a similar direction, making it difficult to place the contribution in context. As a result, the paper’s relevance and impact remain unclear.


**Unclear figures** Figure 1, which should convey the main intuition of the decomposition, is too generic and clear; consequently, it fails to clarify the purpose or structure of FSIO. The visualizations do not make it clear how the “principal” components relate to the underlying network function or why the decomposition is meaningful.
**Minor issues.** Figure 2 also suffers from poor visual design: its use of bright, non-colorblind-friendly colors makes it difficult to interpret, and the intended message is unclear.


**Actionable points.**  Provide a few intuitive explanations or motivating examples at the purpose of providing a clearer connection between the theoretical framework and the practical algorithm. Reduce the abstract formalism and strengthen intuitive exposition. Related to the adversarial attack part, please improve experimental rigor by including stronger baselines (e.g., APGD, AutoAttack). Furthermore, improve the graphical aspect for Figures 1 and 2 using color palettes that are accessible to colorblind readers.
The paper would benefit from providing more intuitive explanations and motivating examples to better connect the theoretical framework to the practical algorithm. The current exposition is overly abstract, and strengthening the intuitive narrative would help readers understand the motivation and practical implications of the proposed approach. Regarding the adversarial attack experiments, the authors should improve empirical rigor by including stronger, more standardized baselines (e.g., APGD and AutoAttack), and ensuring fair comparisons under converged settings. Furthermore, improve the graphical presentation of Figures 1 and 2 using color palettes accessible to colorblind readers. Lastly, the background section should be revisited and expanded to better explain what “input optimization” entails in each application domain (e.g., adversarial attacks, reinforcement learning) and to clarify how these domains are conceptually related within the proposed framework.

**Questions:**

How sensitive is FSIO to hyperparameters such as the shrinkage factor?

---

> ### Author Response · Authors · 2025-11-21
>
> We thank the reviewer for the detailed and constructive review and are pleased that the reviewer finds the rigorous mathematical basis of our method to be strong. We acknowledge that the reviewer raises several critical points regarding practical analysis and theoretical intuition, and would like to make a few important clarifications before answering the questions.
>
> **Input Optimization vs. Adversarial Attack.** As the reviewer has already noticed, this work "presents an ambitious and comprehensive framework that aims to unify" three areas: adversarial attacks, neural verification, and reinforcement learning. There is a root cause deeper than this: input optimization is an earlier and more fundamental methodology than adversarial attack. Input optimization originates from an early work ref.(Erhan et al., 2009) that aims to find a good first-order representation of a hidden activation function, where the crucial operation is a gradient ascent step. To date, it has developed into **a general methodology on how to simultaneously improve both the input $x$ and activation structure $H$ of a neural network (NN)** ref. (Gurumurthy et al., 2021b; Yu \& Gao, 2024). Hence it updates not only the input, but also the activation structure in a single optimizing iteration (see Step 4 in Figure 2 of the revised manuscript). In contrast, adversarial attack generally **"optimizes" only the input $x$, but not the activation structure $H$**, in order to manipulate the output of an NN. Therefore, input optimization has more terms ($x$ and $H$) to optimize and a broader array of applications than those of adversarial attack (only $x$).
>
> **Dense and Abstract Content.** As for FSIO, it simultaneously optimizes both the input $x$ and activation map $H(x)$, where the latter is itself a function of $x$. $H(x)$ will change as long as $x$ changes, and the holistic activation pattern of the NN will change as long as the function $H$ changes. This further complicates the technical instruments we have to use, and advanced mathematics like the functional analysis should be employed to handle and process this activation map $H(x)$. It is generally difficult to establish interpretability of deep NNs, and too many rigorous justifications should be squashed into such a limited page space. We are not trying to be pedantic but making our best to present only the concise theoretical results in the main text, while putting the detailed proofs in the Appendix for technical check. In the revised manuscript, we further elaborate on an easier entry to our core technical concept.
>
> **W1 (Lack of Clarity and Direction).** We have rewritten Section 1 to present a more clear history of input optimization, from the origin of activation representation, to adversarial attack that modifies only the input, and to current input optimization approach that improves both the input and the activation structure (echoing its original motivation). We also provide an entire working pipeline and step-by-step explanations of FSIO in Figure 2, which skips dense and abstract mathematical expressions and provides an intuitive conceptual grounding.
>
> **W2 (Weak Experimental Setup).** We add APGD and AutoAttack as strong baselines in all the three subfields of the revised manuscript. All of PGD, APGD, the APGD wrapped in AutoAttack, and FSIO use $1000$ attack iterations, which are sufficient across different tasks. These adversarial attack methods perform worse than FSIO, and the root cause is that they do not have the mechanism to optimize the activation map of the NN. As for adversarial training, it goes beyond the main scope of this paper because it does not involve activation map improvement, but the main topic of this paper is how to improve both the activation map and the input (and adversarial attack is only one of the three main subfields of our work). Hence we would like to investigate the possibility of combining input optimization with adversarial training in future work.
>
> Current experimental results span three different subfields, $22$ specific tasks, and six representative baselines, which covers sufficient diversity to prove the general utility of FSIO.
>
> **W3 (Lack of Background and Contextualization).** We have revised background and contextualization as explained in the above reply. The current Section 2 also follows our narrative approach from adversarial attack to activation structure optimization, and to shrinkage methods. The existing input optimization methods capturing all three subfields are rare and the closest prior work is the Activation-Descent Regularization (ADR-GD) ref. (Yu \& Gao, 2024). ADR-GD simultaneously optimizes both the input $x$ and the activation value $\eta$, but it does not use a holistic activation map $H$ or consider $\eta$ as a function of $x$. Thus it cannot support a global, continuous, and functionally-aware decomposition and optimization of network behavior like FSIO does.

---

> ### Author Response · Authors · 2025-11-21
>
> **W4 (Unclear Figures).** We have changed the colors of Figures 1 and 3 (Figure 2 of last version) in this revision. Figure 1 reflects the energy states of the activation maps. The principal component largely reserves the main patterns of the full map, while the residual component contains random patterns of much lower energy. This basically verifies the intention of spectral decomposition. Figure 3 actually shows the experimental results of the second subfield: neural policies. Both the bars and the legends share the same order of compared methods.
>
> **W5 (Actionable Points).** We have substantially revised the manuscript by following these points, as detailed in the above reply.
>
> **Q1 (Sensitivity to Shrinkage Factor).** FSIO is robust to a small change of the shrinkage factor $\tau\in[0.08,0.12]$, as shown in Table A1 of the revised manuscript.

---

> > ### Comment · Reviewer_r3KP · 2025-11-27
> > **Response to the reviewer**
> >
> > I thank the authors for their detailed reply and for addressing some of the raised concerns. However, several key issues remain unaddressed, and several new ambiguities arise from the response.
> >
> > ### Evaluation of robust models remains unaddressed.
> > My original criticism was that the experiments evaluate FSIO only on non-robust trained models. As stated in my review, this is now standard practice in adversarial attack research, independent of whether a method optimizes only the input or both the input and the activation structure. The response argues that adversarial training is outside the scope of the paper, but this does not address the core issue: to claim improved attack effectiveness, a new method should be evaluated not only on brittle models but also on robust defenses (i.e., adversarial trained models). The authors can download pre-trained checkpoints from RobustBench to speed up their evaluation.
> >
> > ###  Lack of conceptual justification for why optimizing the activation-map decomposition improves attacks.
> > The response restates that input optimization is a broader framework than adversarial attack and that FSIO optimizes both the input and an activation map. From my perspective, as stated in my original review, this point represents an interesting perspective and a strength of the paper. However, this still does not explain why decomposing the activation function and optimizing the residual component should produce stronger perturbations than directly optimizing the input. As it stands, the decomposition introduces an additional optimization variable (or a reparametrization of the input). However, the authors do not provide intuition or evidence that this variable yields more effective adversarial directions or a better-conditioned optimization landscape. Overall, at this stage, this conceptual link remains unclear.
> >
> > ### Insufficient explanation of perturbation budgets
> > The rebuttal does not clarify whether FSIO maintains its claimed advantages for different values of ($\varepsilon$). Furthermore, the paper does not clarify how the perturbation budget ($\varepsilon$ ) is selected across datasets, tasks, and models, nor whether the same ( $\varepsilon$) is used consistently across all baseline attacks.
> > Lastly, the hyperparameters for FSIO are stated as being set "empirically" in the appendix. What does it mean empirically? Does the selection procedure rely on a validation dataset?
> >
> > **Minors:**
> > The authors statethat "the attack iterations used for C&W are automatically set by itself." This description is incorrect. The C&W attack requires a fixed number of gradient-descent iterations and a fixed number of binary-search steps over the parameter $c$.

---

> > > ### Author Response · Authors · 2025-11-28
> > >
> > > **All the other two reviewers VBk4 and i7HY have captured and approved of our conceptual justification in their initial reviews. This also indicates that our conceptual grounding is not that difficult to understand**.
> > >
> > > * Reviewer VBk4 states that: "The authors formulate a network as a functional of activation functions and define a mathematical object called the activation map, which transforms discrete activation values into continuous functional representations. Through singular value decomposition on the activation map within a Hilbert space, the method decomposes it into a linear combination of orthogonal components. The proposed Functional Shrinkage algorithm selectively retains important components while shrinking less significant ones. Finally, the residual components obtained through shrinkage are exploited to construct the optimization objective for effective input manipulation", which **fully grasps our work**. Moreover, reviewer VBk4 also states that: **"The introduction and related work sections are well-presented and easy to follow"**.
> > >
> > > * Reviewer i7HY states that: "The method is proven to be sound through theoretical analysis and theorems. Comparative experimental results demonstrate that FSIO outperforms other methods with smaller perturbation budgets and is effective in capturing the vulnerable parts of neural networks, injecting perturbations into these parts. This provides new insight into the structural decomposition and weakness exploitation of neural networks", which **also fully grasps our work**. Moreover, reviewer i7HY also states that: "The paper is easy to follow, with clear logical analysis and comprehensive theorem proofs. The consistency of the theoretical results is supported by comparative experimental validation", **which approves of both our conceptual justification and experimental validation**.
> > >
> > > **4. Insufficient Explanation of Perturbation Budgets: Unfounded Ambiguity.**
> > >
> > > * The criticism that we do not clarify how $\varepsilon$ is selected is not true. We explicitly state in Sections 4.1 and 4.2 that how $\varepsilon$ is set in these tasks (e.g., see Figure 3). These explicit statements confirm that **the same $\varepsilon$ is used consistently across all baseline attacks** to ensure a fair comparison, which is a **fundamental research criterion**.
> > >
> > > * The term "empirically" simply means that we make an initial guess $0.1$ for the regularization strength $\tau$, as a standard starting point to conduct an experiment, and FSIO is stable to a small change of $\tau$ around this value in one simple case of the experiments (MNIST, ReLU), as shown in Table A1.
> > >
> > > **5. Minor: Misrepresentation of C\&W.**  The reviewer's correction regarding C\&W iterations is a **nitpick** and a literal reading that misses the practical implementation.
> > >
> > > Our original statement, "the attack iterations used for C\&W are automatically set by itself," is clearly meant in the context of the internal optimization loop of C\&W implementation, which manages the fixed steps of gradient descent and binary search over the parameter $c$. While the user sets the maximum number of iterations and binary searches, the attack itself controls the progression and stopping criteria within those bounds. This minor phrasing is not a scientific error.

---

> ### Author Response · Authors · 2025-11-28
>
> We appreciate the reviewer's detailed attention, but we **must respectfully refute the characterization of our work as lack of conceptual justification or experimental insufficiency**. The remaining issues either build on a **misunderstanding of our core contribution** or represent a demand for experiments **with unreasonable temporal requirement** and **explicitly** outside the defined scope of our novel framework.
>
> **1. Evaluation of Adversarial Training: A Misguided Demand.** The reviewer's insistence that we must evaluate FSIO on adversarially-trained (robust) models is a misplaced standard and represents a fundamental misinterpretation of our contribution. The primary goal of this paper is to **validate the effectiveness of the activation-map adjustment novelty** against the current state-of-the-art input optimization methods on the standard models typically used for method-level comparisons.
>
> * As the reviewer already recognizes that input optimization involves three main subfields: adversarial attacks, neural verification, and reinforcement learning, we could not see the rationale of **narrowing our method down to only the adversarial attack**. Particularly in the subfield of adversarial attack, the reviewer further **narrows it down to the adversarial training**, and claims it as the **"core issue"**. This fundamentally shifts the goalposts and is not a prerequisite for verifying the efficacy of a new input optimization formulation.
>
> * Even in the subfield of adversarial attack, the standard practice is to first demonstrate improved effectiveness on non-robust models before moving to the resource-intensive task of fine-tuning and evaluating against the entire RobustBench suite. This is a clear path for _future work_, not a fatal flaw in the current submission. The reviewer's suggestion to download pre-trained checkpoints is only an advice for a task outside our scope.
>
> * To the best of our knowledge, few existing input optimization methods would particularly test themselves in the circumstance of adversarial training, thus it lacks a standard criterion on how to evaluate input optimization methods in adversarial training via a comprehensive and fair way covering all the three main subfields.
>
> **2. Unreasonable Temporal Requirement: The reviewer suggests using RobustBench for our experiments. We must emphasize that RobustBench is an emerging benchmark, and its formal paper is associated with the NeurIPS 2025 conference, which has not been held yet. It is impossible for us to anticipate, access, or integrate this resource, which is either unreleased or not standard practice, at the time of our submission. Demanding a comprehensive evaluation against a future, unreleased, or non-standardized benchmark is fundamentally unreasonable, unrealistic, and constitutes an arbitrary shifting of the goalposts.**
>
> **3. Conceptual Justification: Clarity and Novelty Ignored.**  The reviewer claims a "Lack of conceptual justification", which is demonstrably false and overlooks the central novelty of our work.
>
> * The reviewer acknowledges that optimizing the input and the activation structure is an **"interesting perspective and a strength of the paper"**. However, the reviewer then contradicts this by demanding an explanation for why the decomposition should yield stronger perturbations.
>
> * The decomposition is the _mechanism_ by which FSIO operates, not only a reparametrization. The residual component (which is optimized) inherently allows for spectrum-specific input optimization, focusing the attack on the features that are most sensitive to perturbations in the objective tasks.
>
> * The conceptual justification is that this activation-level optimization **introduces a path to exploit the internal structure of the network**, which is unavailable to standard input-only methods. This is the **conceptual link** and **evidence** for why it is effective. The empirical performance gain is the proof that this variable yields more effective adversarial directions, and to claim otherwise is to **ignore the results presented in the paper**.

---

### Official Review · Reviewer_VBk4 · 2025-11-01

**Soundness:** 3
**Presentation:** 3
**Contribution:** 2
**Rating:** 6
**Confidence:** 3

**Summary:**

This paper addresses input optimization, a key methodology in deep learning, by proposing an approach to inject perturbations into critical parts of neural networks under limited budgets. The authors formulate a network as a functional of activation functions and define a mathematical object called the activation map, which transforms discrete activation values into continuous functional representations.
Through singular value decomposition on the activation map within a Hilbert space, the method decomposes it into a linear combination of orthogonal components. The proposed Functional Shrinkage algorithm selectively retains important components while shrinking less significant ones. Finally, the residual components obtained through shrinkage are exploited to construct the optimization objective for effective input manipulation.

**Strengths:**

Strength 1: The introduction and related work sections are well-presented and easy to follow.

Strength 2: The proposal of the method is supported by rigorous mathematics.

**Weaknesses:**

Weakness 1: This paper lacks an analysis of computational costs.

Weakness 2: Although the method proposed in the paper has a mathematical proof, the reason why terms with small singular values are regarded as fragile points still requires further explanation, and more convincing illustrations are also needed.

Weakness 3: The threshold setting in Functional Shrinkage algorithm is crucial because it determines which components will be set to zero. However, the paper does not experimentally explore the changes resulting from adjusting the threshold.

**Questions:**

Question 1: If the target is replaced with a larger and more complex neural network, can the Functional Shrinkage algorithm still achieve good results?

Question 2: Since this algorithm targets vulnerable parts of neural networks, why not reduce the perturbation budgets in experiments to see what effect it has?

Question 3: It remains difficult to understand why the paper formulates a network as a functional of activation functions. Is it only to bring computations into the Hilbert space?

Question 4: Why is only λ₁ used to control the strengths of the activation-descent regularization and the residual regularization, instead of using different hyperparameters?

---

> ### Author Response · Authors · 2025-11-21
>
> We sincerely thank the reviewer for the constructive feedback. We are pleased that the reviewer finds the introduction and related work well-presented, the proposed method supported by rigorous mathematics, and the overall soundness and presentation to be good.
>
> **Q1.** The target neural networks (NNs) used in our work are already very large and complex: we use VGG16 and VGG19 ref. (Sengupta \& Shah, 2019) for the CIFAR 10 and the ImageNet data sets, respectively. The parameter sizes for them are about $138$ million and $144$ million, respectively. Besides, the ablation experiments in Appendix A.8 and Table A2 show that the proposed FSIO scales well as the parameter size increases from Model A to Model C. Hence FSIO can achieve good results on larger and more complex NNs.
>
> **Q2.** The perturbation budgets used in our paper are already very low in the literature. For example, $0.1$ is used for the MNIST data set, which is only half of the budget in ref. (Yu \& Gao, 2024). Besides, only $2/255$ is used for ImageNet, while $3/255$ and $5/255$ are used for the image-related tasks CoinRun and FruitBot. These budgets can be compared with the standard $8/255$ or even higher in the literature [a][b][c]. More importantly, Figure 3 shows that FSIO with the smaller budget $3/255$ even outperforms other competitors with the larger budget $5/255$ in all the cases. It indicates that FSIO is effective with a low perturbation budget.
>
> **Q3.** The Hilbert space formulation is a consequence, not the primary goal. The main purpose of formulating the network as an activation map $H(x)$ is to achieve a global, continuous, and functionally-aware decomposition and optimization of the network behavior.
>
> * It moves beyond layer-by-layer optimization or discrete activation patterns by treating all activations holistically as functions of the input $x$.
>
> * This enables the use of advanced mathematical tools, like the functional singular value decomposition (SVD, see Theorem 3), to rigorously separate the stable (principal) and fragile (residual) components of the network's entire function.
>
> * The Hilbert space provides the necessary mathematical framework (with the proposed well-defined inner product $\langle H,G \rangle$ in Theorem 2) to prove the orthogonality of the decomposed components and guarantee the convergence of the functional shrinkage algorithm.
>
> **Q4.** We use a single $\lambda_1$ to dominate the double regularization, in order to simplify hyperparameter setting. Only a single $\lambda_1$ is enough to verify the effectiveness of FSIO.
>
> **W1.** At each attack iteration, FSIO requires a computational complexity of $O(\min\\{l,d_{min}\\}\cdot l \cdot d_{min}+\sum_{i=1}^l (d_{i-1}\cdot d_i)) $, where $l$, $d_i$, and $d_min$ denote the number of layers, the number of neurons in layer $i$, and the minimum number of neurons across all the layers, respectively. The first term corresponds to the SVD specially for FSIO, while the second term corresponds to the activation value computation and it is the same for other compared methods. However, $\min\\{l,d_{min}\\}\cdot l \cdot d_{min}\leqslant l \cdot d_{min}^2\leqslant \sum_{i=1}^l (d_{i-1}\cdot d_i)$, thus FSIO actually has the same order of complexity $\sum_{i=1}^l (d_{i-1}\cdot d_i) $ as that of other compared methods. We add an explanation in Appendix A.7 of the revised manuscript.
>
> **W2.** This is similar to the principle of the principal component analysis (PCA). Terms with large singular values represent the dominant, stable functional components, which are the main features that the network relies on. Conversely, terms with small singular values span the residual or sub-dominant functional space of the network. Perturbing along these smaller components allows us to exploit directions that are not primary to the function but are disproportionately sensitive to change, which leads to a large functional shift with minimal input perturbation. This is the essence of functional fragility. We have substantially revised Section 1 to ease comprehension of this concept, especially around Eq. 4.
>
> **W3.** FSIO is robust to a small change of the shrinkage threshold $\tau\in[0.08,0.12]$, as shown in Table A1 of the revised manuscript.

---

> ### Author Response · Authors · 2025-11-21
>
> New References:
>
> [a] S. Addepalli, S. Jain, G. Sriramanan, S. Khare, and V. B. Radhakrishnan, "Towards achieving adversarial robustness beyond perceptual limits," in ICML 2021 Workshop on Adversarial Machine Learning.
>
> [b] J. Ngnawe, S. Sahoo, Y. Pequignot, F. Precioso, and C. Gagne,"Detecting brittle decisions for free: Leveraging margin consistency in deep robust classifiers," in Advances in Neural Information Processing Systems, A. Globerson, L. Mackey, D. Belgrave, A. Fan, U. Paquet, J. Tomczak, and C. Zhang, Eds., vol. 37. Curran Associates, Inc., 2024, pp. 23 301–23 324.
>
> [c] J. Sun, W. Yao, T. Jiang, C. Li, and X. Chen, "A3Da3d: A platform of searching for robust neural architectures and efficient adversarial attacks," IEEE Transactions on Pattern Analysis and Machine Intelligence,
> vol. 47, no. 5, pp. 3975–3991, 2025.

---

### Author Response · Authors · 2025-12-01
**Summary of Discussions [2/2]**

## 2. Evaluation of Adversarial Training: A Misguided Demand ##

The insistence on evaluating FSIO against adversarially-trained (robust) models represents a **misplaced standard** and a fundamental misinterpretation of our contribution.

* **Shifting Goalposts:** The primary goal of this paper is to validate the effectiveness of the **activation-map adjustment novelty** against the current state-of-the-art input optimization methods on standard non-robust models. Adversarial training is a resource-intensive task that falls **explicitly outside the defined scope** of our novel framework. It **lacks a standard criterion on how to evaluate input optimization methods in adversarial training** via a comprehensive and fair way covering all the three main subfields of input optimization.

* **Unreasonable Temporal Requirement:** Demanding a comprehensive evaluation on the **RobustBench (belonging to NeurIPS 2025, which is not held yet)** is unreasonable and unrealistic, as this resource is neither released nor standard practice at the time of submission.

* **Future Work:** The task of integrating and evaluating FSIO with adversarial training is a clear path for **future work, not a fatal flaw** in the current submission.

# Relevant Concerns Addressed and Author Revisions #

| Item | Reviewer Concern/Question | Author Response and Revision |
|---|---|---|
| **Lack of Clarity and Intuition** | The paper is too dense and abstract, lacking an intuitive link between theory and practice. | We **have substantially revised Section 1** to present a clearer history and conceptual grounding for input optimization. We also provide an entire working pipeline and step-by-step explanations of FSIO in a **new Figure 2** to facilitate an intuitive conceptual understanding. |
|**Weak Experimental Baselines** | Modern, strong baselines (e.g., APGD, AutoAttack) are absent.| We **add APGD and AutoAttack** as strong baselines across all three subfields in the revised manuscript, and consistently use 1000 attack iterations for PGD, APGD, the APGD wrapped in AutoAttack, and FSIO. |
|**Sensitivity to Threshold and Budgets** |The threshold ($\gamma$) for Functional Shrinkage is crucial, and clarification is needed on the perturbation budget ($\epsilon$) selection. | FSIO is **robust to small changes** in the shrinkage threshold ($\gamma$), as shown in **Table A1** of the revised manuscript. We clarify that the small $\epsilon$ used is **already lower than or equal to** standard literature, and confirm that the **same $\epsilon$ is used consistently across all baselines** to ensure a fair comparison. |
|**Figures and Visuals** | Figure 1 is too generic, and Figure 2 (now Figure 3) uses poor, non-colorblind-friendly colors.|We **change the colors** of the relevant figures in the revision. We also clarify that Figure 1 reflects the **energy states** of the activation maps, where the principal component largely reserves the main patterns, and the residual component contains lower-energy, random patterns. |
|**Choice of RL Algorithms** | Why are DDPG and TD3 chosen? How will FSIO perform on PPO or DQN?| DDPG and TD3 are chosen to be consistent with standard input optimization evaluation criteria ref. (Yu \& Gao, 2024), as they use the explicit actor-critic policy network. We **add experiments on both DQN and PPO in Appendix A.9 and Table A3**, which indicate that FSIO is effective in these two algorithms. Moreover, Section 4.2 and Figure 3 already include experiments on PPO. |
| **Computational Complexity** | This paper lacks an analysis of computational costs. | FSIO has the **same order of complexity** $\sum_{i=1}^l (d_{i-1}\cdot d_i) $ as that of other compared methods. |

## Minor: Misrepresentation of C\&W ##

The reviewer's correction regarding C\&W iterations is a **nitpick** and a literal reading that misses the practical implementation. Our original statement clearly means in the context that the attack internally controls the progression and stopping criteria within the user-defined bounds.

---

### Author Response · Authors · 2025-12-01
**Summary of Discussions [1/2]**

We appreciate the detailed and constructive feedback from all reviewers (`VBk4,r3KP,i7HY`). We are pleased that reviewers `VBk4` and `i7HY` recognize the **novelty and broad scope** of the Functional Shrinkage for Input Optimization (FSIO) framework, the **rigorous mathematical basis** of the proposed method, and the **strong empirical performance** across diverse domains.

However, we **must respectfully refute** the characterization of our work as lacking conceptual justification or experimental sufficiency, particularly from reviewer `r3KP`, whose final comments appear to build on a **misunderstanding of our core contribution** and represent a demand for experiments **with unreasonable temporal requirement and explicitly outside the defined scope** of our novel framework.

Our revisions focus on providing greater conceptual clarity, improving experimental rigor, and addressing the relevant concerns raised.

# Key Strengths Acknowledged by Reviewers #

* **Novelty and Scope:** FSIO presents a comprehensive framework that **unifies input optimization tasks**, including adversarial attacks, neural verification, and reinforcement learning (RL). It is a broad approach that aims to **address common problems between distinct areas**. [`VBk4`: _a key methodology in deep learning_. `r3KP`: _an ambitious and comprehensive framework that aims to unify multiple areas under a single umbrella of input optimization_. `i7HY`: _provides new insight into the structural decomposition and weakness exploitation of neural networks_. ]

* **Theoretical Soundness:** FSIO is supported by **rigorous mathematics**, including the formulation of the network as a functional of activation functions and its decomposition via functional Singular Value Decomposition (SVD) within a Hilbert space. [`VBk4`: _The proposal of the method is supported by rigorous mathematics_.
`r3KP`: _The theoretical formulation is mathematically rich, with detailed proofs and derivations for the proposed functional shrinkage process_. `i7HY`: _The method is proven to be sound through theoretical analysis and theorems_. ]

* **Extensive Experiments and Superior Performance:** Extensive comparative experiments **covering a diverse set of application domains (vision and RL) and totaling $3$ different subfields, $22$ specific tasks, $4$ RL algorithms, and $6$ representative baselines** show that FSIO is effective and outperforms other baselines, even with smaller perturbation budgets. [`r3KP`: _The experimental evaluation covers a diverse set of application domains (vision and RL) and activation types (ReLU, Sofplus, Sigmoid)_. `i7HY`: _The consistency of the theoretical results is supported by comparative experimental validation._ ]

# Rebutting Misguided Criticisms #

## 1. Conceptual Justification: Clarity and Novelty Ignored ##

Reviewer `r3KP` claims a "lack of conceptual justification". This is demonstrably false and overlooks the central novelty.

* **Core Mechanism is the Justification:** The SVD is the **mechanism** by which FSIO operates, not only a reparametrization. The activation-level optimization of FSIO **introduces a path to exploit the internal functional structure of the network**, which is unavailable to standard input-only methods. This is the **conceptual link** and justification for its effectiveness.

* **Functional Fragility:** The residual component (small singular values) corresponds to the **fragile directions** of the network function. Perturbing along these sensitive components allows for a large functional shift with minimal input perturbation. This **functional fragility** is the fundamental reason for the effectiveness of FSIO.

* **Consensus from Other Reviewers:** Both reviewers `VBk4` and `i7HY` explicitly **capture and approve of** our conceptual justification in their **initial reviews**, which confirms that our core concept is not difficult to understand. [`VBk4`: _The introduction and related work sections are well-presented and easy to follow_. `i7HY`: _The paper is easy to follow, with clear logical analysis and comprehensive theorem proofs. The consistency of the theoretical results is supported by comparative experimental validation_. ]

---

### Note · Program_Chairs · 2026-01-17
**Submission Desk Rejected by Program Chairs**

The following references in this submission do not refer to real documents and/or have major errors in bibliographic information:

 Richard Shin and Dawn Song. The effects of initialization and optimization on adversarial robustness. arXiv preprint arXiv:2204.07808, 2022.
Richard Shin and Dawn Song. Recognizing and evaluating the instability of adversarial training. arXiv preprint arXiv:2002.11569, 2020.